# COLLABORATION OF EXPERTS: ACHIEVING 80% TOP-1 ACCURACY ON IMAGENET WITH 100M FLOPS

## ABSTRACT

In this paper, we propose a Collaboration of Experts (CoE) framework to assemble the expertise of multiple networks towards a common goal. Each expert is an individual network with expertise on a unique portion of the dataset, contributing to the collective capacity. Given a sample, delegator selects an expert and simultaneously outputs a rough prediction to trigger potential early termination. For each model in CoE, we propose a novel training algorithm with three components: weight generation module (WGM), label generation module (LGM) and selection reweighting module (SRM). WGM adapts the losses of experts, enabling them to focus on different portions of the dataset. LGM generates the label to constitute the loss of delegator for expert selection. SRM aims to promote delegator to select experts better. CoE achieves the state-of-the-art performance on ImageNet, 80.7% top-1 accuracy with 194M FLOPs. Combined with PWLU activation function and CondConv, CoE further boosts the accuracy to 80.0% with only 100M FLOPs for the first time. Furthermore, experimental results on the translation task also demonstrate the strong generalizability of CoE. CoE is hardware-friendly, yielding a 3∼6x acceleration compared with existing conditional computation approaches.

## 1 INTRODUCTION

There are many approaches for model collaboration, among which ensemble learning (Hansen & Salamon, 1990; Wen et al., 2020; Wenzel et al., 2020) is a popular one. Ensemble learning uses a consensus scheme to decide the collective result by vote. However, it requires multiple forward passes, leading to a significant runtime cost. MIMO (Havasi et al., 2021) draws inspiration from model sparsity (Frankle & Carbin, 2019) and tries to ensemble several subnetworks within one regular network. It only needs one single forward pass of the regular network but is incompatible with compact models. Conditional computation methods (Cheng et al., 2020; Yan et al., 2015; Shazeer et al., 2017) alleviates this issue via delegation scheme, i.e. assigning one or several, rather than all models, conditionally to make the prediction. Some recently proposed conditional computation methods (Zhang et al., 2020b; Yang et al., 2019; Zhang et al., 2021) have achieved remarkable performance based on dynamic convolution. Nonetheless, they usually have high memory access cost (MAC) and a low degree of parallelism, which increases the real latency (Ma et al., 2018).

Motivated by this, we propose the Collaboration of Experts (CoE) framework to both eliminate the need for multiple forward passes and keep hardware-friendly. CoE consists of one delegator and multiple experts. Firstly, delegator gives a rough prediction and makes the expert selection. If the rough prediction is unreliable, the selected expert will make the refined prediction. Otherwise, the procedure will be early terminated to save FLOPs. Moreover, we only need to load the selected expert into memory, thus keep the ratio of MAC to FLOPs as a constant. By contrast, dynamic convolution methods (Zhang et al., 2020b; 2021) need to load a large number of parameters, namely basis models or experts, to synthesize the input-dependent ones. It enlarges MAC and reduces the degree of parallelism, resulting in a significant deceleration.

To make each model in CoE play its role, we propose a novel training algorithm (as shown in Fig.1) which consists of three components: weight generation module (WGM), label generation module (LGM) and selection reweighting module (SRM). LGM generates the label (selection label) to constitute loss of delegator for expert selection (selection loss). Selection label is a one-hot vector, indicating the suitable expert for each given input. Due to the random initialization of experts, selection labels are irregular in the early training stage. Nonetheless, delegator tends to learn generalizable patterns first, since networks learn gradually more complex hypotheses during

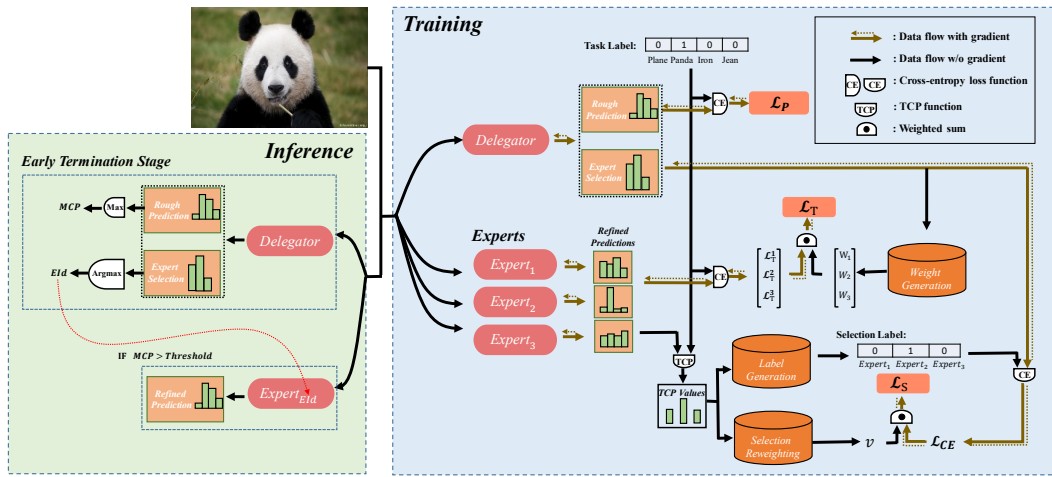

Figure 1: Framework of CoE with 3 experts. Delegator outputs the probabilities to select each expert and a rough prediction to trigger potential early termination. Each input is processed through all experts during training, but only through at most one expert during inference.

training (Arpit et al., 2017). With delegator as the bridge, WGM can partition the training data into portions based on generalizable patterns, and reweight expert losses to impel each expert to focus on one portion. This procedure makes selection labels more regular in return. Thus, delegator avoids overfitting to the irregular labels. SRM aims to promote delegator to select experts better. It is achieved by enforcing delegator to focus on samples whose recognition results are sensitive to expert selection.

Experimental results on ImageNet demonstrate the superiority of CoE. It achieves 78.2/80.7% top-1 accuracy with only 100/194M FLOPs, while the accuracy for ensembled models (Hansen & Salamon, 1990) is 79.6% with 920M FLOPs. Compared with dynamic network approaches, CoE is more hardware-friendly. It not only outperforms the SOTA dynamic method BasisNet which achieves 80.0% accuracy with 198M FLOPs (Zhang et al., 2021), but also accomplishes a 3.1x speedup on hardware. Besides, CoE can be equipped with CondConv and further improve the accuracy to 79.2/81.5% with 102/214M FLOPs. More surprisingly, we further boost the accuracy to 80.0% with only 100M FLOPs for the first time by using PWLU activation function (Zhou et al., 2021). Experimental results on the translation task also demonstrate the strong generalizability of CoE.

The contributions of this paper can be summarized as follows:

- We propose a collaboration framework named Collaboration of Experts (CoE) and demonstrate that it can lead to outstanding performance with little computation cost. Moreover, it is hardware-friendly and able to achieve real speedup.
- We present a novel optimization strategy for CoE. The core insight is to promote diversity within experts by distributing their expertise over different portions of the dataset.
- We update the state-of-the-art on ImageNet for mobile setting, achieving 80.0% top-1 accuracy with only 100M FLOPs for the first time.

## 2 RELATED WORK

### 2.1 ENSEMBLE LEARNING AND MODEL SELECTION

Ensemble learning (Hansen & Salamon, 1990) aims at combining the predictions from several models to get a more robust one. Some recently proposed literatures (Wen et al., 2020; Wenzel et al., 2020) demonstrate that significant gains can be achieved with negligible additional parameters compared to the original model. However, these methods still require multiple (typically, 4-10) forward passes for prediction, leading to a significant runtime cost. Differently, CoE utilizes a delegator to select only one expert for the refined prediction, thus at most two forward passes are needed. MIMO (Havasi et al., 2021) draws inspiration from model sparsity (Frankle & Carbin, 2019) and holds the view that multiple independent subnetworks can be concurrently trained within one regular network because of the heavy parameter redundancy. Therefore, those subnetworks can be ensembled with a single forward pass of the regular model. However, MIMO cannot be applied to compact models which

have already been pruned or the ones constructed by AutoML methods (Zhong et al., 2018; Zhang et al., 2020a; Cai et al., 2020). It is because these models are compact enough and have few redundant parameters. By contrast, CoE is free from the compactness of experts and compatible with various models. Recently, some works about model selection are proposed (Li et al., 2021b; You et al., 2021). These methods are concerned with ranking a number of pre-trained models and finding the one transfers best to a downstream task of interest. Therefore, they select models task-wisely. By contrast, CoE aims at improving the task performance via selecting the most suitable expert for each sample instance-wisely. Moreover, these methods conduct model selection based on a set of samples (training set) which cannot be adopted in CoE.

## 2.2 DYNAMIC NETWORKS

Dynamic networks achieve high performance with low computation cost by conditionally varying the network parameters (Zhang et al., 2020b; Yang et al., 2019) or network architectures (Yuan et al., 2020). HD-CNN (Yan et al., 2015) and HydraNet (Mullapudi et al., 2018) select branches based on the category, they cluster all categories into n groups, where n is the number of branches. While CoE learns the model selection pattern automatically, it can be based on any property, rather than limited to the category. MoE (Shazeer et al., 2017) and Switch Transformer (Fedus et al., 2021) enable the direct training of Router by scaling the output feature of experts with the predicted gate-values of Router. These methods aim at conditionally selecting a specific layer or block. Differently, CoE can take more advantage of conditional computation as the selection of whole network makes every parameter input-dependent. The recently proposed Dynamic Convolution methods (Zhang et al., 2020b; Yang et al., 2019; Chen et al., 2020) share the same idea and achieve remarkable performance with low FLOPs but high latency. It is because these methods need to load many basis models or experts to synthesize the dynamic parameters, causing high MAC and low degree of parallelism (Ma et al., 2018). By contrast, CoE only needs to load the selected expert into memory, avoiding these problems. Finally yet importantly, batch processing is an important method to enhance the degree of parallelism. Because of the input-dependent parameters (Zhang et al., 2021) or architectures (Yuan et al., 2020), these methods cannot process samples in batch. Differently, CoE supports batch processing because the number of experts is limited and each one of them corresponds to many test samples.

## 3 METHOD

As shown in Fig.1, CoE consists of a delegator and $n$ experts, a total of $n + 1$ individual neural networks. To make the best of limited capacity, each expert is encouraged to focus on just one unique portion of the dataset. This is achieved by WGM which reweights training losses of experts. Given a sample, delegator selects an expert and simultaneously outputs a rough prediction to trigger potential early termination. It is trained with cross-entropy loss and labels generated from LGM that indicate the suitable expert. Moreover, SRM aims to boost delegator by enforcing it to focus on samples whose recognition results are sensitive to expert selection. Since the inference of delegator is conducted all the time, we prefer to make delegator more lightweight than expert. Delegator consists of three modules: feature extractor, task predictor and expert selector as shown in Fig.2. Based on the feature derived from feature extractor, task predictor and expert selector output the probabilities for classification and expert selection respectively. We describe the inference procedure and training strategy of CoE comprehensively in the following subsections. The number of samples and experts are denoted as $m$ and $n$ respectively.

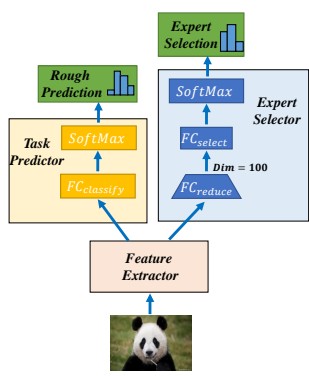

Figure 2: Architecture of delegator.

## 3.1 INFERENCE PROCEDUREF

CoE firstly uses delegator to obtain the rough prediction and determine the selected expert for each sample. Afterward, *Maximum Class Probability* (MCP, Corbière et al. 2019) of rough prediction is calculated. It is the probability of the predicted class. For samples with MCP larger than a given threshold $\tau$, the final recognition result is derived from the rough prediction (early termination). Other samples are partitioned into $n$ groups based on which expert is selected. Then batch processing can be conducted within each group to obtain the refined prediction. The averaged FLOPs/Instance of CoE ranges from $F_D$ to $F_D + F_E$ by varying $\tau$, $F_D$ and $F_E$ are FLOPs of delegator and experts.

## 3.2 LABEL GENERATION MODULE (LGM)

LGM generates the label (selection label) to constitute loss of delegator for expert selection (selection loss). Selection label is a one-hot vector, indicating the suitable expert for a given input. Model accuracy can be measured by *True Class Probability* (TCP, Corbière et al. 2019):

$$\text{TCP}_{j,k} = P(Y = y_j | x_j, \text{Expert}_k), \tag{1}$$

where, $x_j$ is the j-th sample, $Y$ and $y_j$ are the predicted and true class. But accuracy is not the only factor for suitability. For example, when models are of different sizes, the larger one usually has a higher TCP. But it may not be more suitable, due to the large inference cost. Given that our concern is not the optimization of network architecture, we can suppose no expert is superior to others (*No Superiority Assumption*, NSA). Motivated by NSA, we leverage the standardized TCP as the metric for sutability:

$$S_{j,k} = \frac{\text{TCP}_{j,k} - Mean(\text{TCP}_{:,k})}{Std(\text{TCP}_{:,k})}, \tag{2}$$

where, $\text{Mean}(\text{TCP}_{:,k})$ and $\text{Std}(\text{TCP}_{:,k})$ are mean value and standard deviation for TCPs of $\text{Expert}_k$.

Given $m$ samples and $n$ experts, selection labels can be denoted by a binary matrix $\boldsymbol{L}_{m \times n}$, thus each row of which is a selection label. According to NSA, the sum of each colum vector of $\boldsymbol{L}_{m \times n}$ should be same, namely $\sum_j L_{j,k} = \frac{m}{n}$ for $k = 1, ..., n$. Therefore, $\boldsymbol{L}_{m \times n}$ can be obtatined by maximizing the sum of standardized TCP (i.e. $S_{j,k}$ in Eq.2):

$$\begin{aligned} \min \quad & \sum_{j,k} -S_{j,k} * L_{j,k} \\ s.t. \quad & L_{j,k} \in \{0,1\}, \quad \sum_k L_{j,k} = 1, \quad \sum_j L_{j,k} = \frac{m}{n} \end{aligned} \tag{3}$$

This problem can be modeled as the balanced transportation problem (BTP, Shore 1970), where each sample is a supply source with a supply of one, each expert is a demand source with a demand of $m/n$. $-S_{j,k}$ is the per-unit transportation cost from the j-th supply source to the k-th demand source. We solve this problem via Vogel approximation method (VAM, Shore 1970) as introduced in Appendix A.1, which is a short-cut approach to invariably obtain a good solution.

## 3.3 WEIGHT GENERATION MODULE (WGM)

To maximize the collective capacity of CoE, the dataset needs to be partitioned into portions then WGM encourages each expert to focus on one portion by reweighting losses of experts. The partition can be indicated by an assignment matrix $\boldsymbol{A}_{m \times n}$, with one-hot row vectors. $A_{j,k} = 1$ means the j-th sample $x_j$ is assigned to the k-th expert, thus the loss weight for $\text{Expert}_k$ gets larger than other experts on $x_j$.

A naive partition can be based on expert suitability, namely, partitioning the dataset with selection labels $\boldsymbol{L}_{m \times n}$ generated from LGM. However, it results in a poor generalization to delegator. Assuming $\text{Expert}_k$ is suitable on a sample $x_j$, thus $A_{j,k} = L_{j,k} = 1$. Due to $A_{j,k} = 1$, the loss weight for $\text{Expert}_k$ gets larger than other experts on $x_j$, making $\text{Expert}_k$ more suitable in return. Therefore selection labels cannot be updated. Moreover, selection labels are irregular in the early training stage because of the random initialization, thus selection labels will remain irregular consistently. With the training going on, delegator will overfit to those irregular labels, yielding poor generalization as shown in Fig.3a. As a result, CoE performs poorly because delegator can hardly select a suitable expert during validation. This is also verified in Appendix B.4.3.

Since networks learn gradually more complex hypotheses during training (Arpit et al., 2017), delegator tends to learn generalizable patterns first. Therefore, the partition can be based on generalizable patterns with delegator as the bridge. In this way, selection labels get more regular in return thanks to the reweighting of expert losses in WGM. As shown in Fig.3b, delegator avoids overfitting to the irregular labels, hence generalizing well to the validation set.

Given $m$ samples and $n$ experts, delegator outputs a probability matrix $\boldsymbol{P}_{m \times n} \in \boldsymbol{R}^{m \times n}$, whose element $P_{j,k} \in [0,1]$ represents the probability of assigning the j-th sample to the k-th expert. As analyzed above, it is better to partition the training data based on $\boldsymbol{P}_{m \times n}$, thus $\boldsymbol{A}_{m \times n}$ can be obtained by maximizing $\sum_{j,k} P_{j,k} * A_{j,k}$. Moreover, according to NSA, the number of samples assigned to each expert should be same, i.e. $\boldsymbol{A}_{m \times n}$ needs to satisfy $\sum_j A_{j,k} = m/n$. Thus, $\boldsymbol{A}_{m \times n}$ is optimized by:

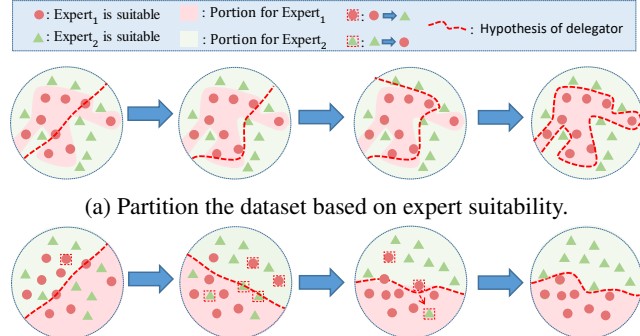

(a) Partition the dataset based on expert suitability.

(b) Partition the dataset based on delegator.

Figure 3: A demo to illustrate the training procedure of CoE. The training samples are denoted as green triangles or pink circles based on which expert is suitable. They are partitioned into two portions based on expert suitability or hypothesis learned by delegator, then WGM enables each expert to focus on one portion. Delegator is trained with selection labels that indicate the suitable expert, thus delegator learns hypothesis to select the suitable expert for each sample.

$$\min \quad \sum_{j,k} -P_{j,k} * A_{j,k}$$

$$s.t. \quad A_{j,k} \in \{0,1\}, \quad \sum_{k} A_{j,k} = 1, \quad \sum_{j} A_{j,k} = \frac{m}{n} \tag{4}$$

This problem can also be modeled as BTP, and solved via VAM as described in section 3.2. In the early training stage, experts are underfitted. Thus we cannot trust $\boldsymbol{A}_{m \times n}$ and need to make the gap between loss weights for experts smaller. We achieve this by smoothing $\boldsymbol{A}_{m \times n}$ to $\overline{\boldsymbol{A}}_{m \times n}$ with Eq.5, where $\alpha$ grows linearly from 0.2 to 0.8 with the training going on,

$$\overline{A}_{j,k} = \begin{cases} \alpha + \frac{1-\alpha}{n}, & if \ A_{j,k} = 1 \\ \frac{1-\alpha}{n}, & if \ A_{j,k} = 0 \end{cases} . \tag{5}$$

Finally, the output of WGM (i.e. $\boldsymbol{W}_{m \times n}$) is obtained by normalizing $\overline{\boldsymbol{A}}_{m \times n}$ with the coefficient $\mathcal{Z} = \sum_{j} \overline{A}_{j,k} = \frac{m}{n}$:

$$W_{j,k} = \frac{\overline{A}_{j,k}}{\mathcal{Z}}. \tag{6}$$

### 3.4 SELECTION REWEIGHTING MODULE (SRM)

Expert suitability can be measured with standardized TCP (Eq.2). If experts have similar suitabilities for a given sample, the expert selection will have little influence on the final performance of CoE. As the capacity of delegator is limited, it should pay less attention to those samples. This is achieved by SRM, which reweights losses of delegator based on a statistic that reflects suitability similarity.

The output of SRM is denoted as $\{v_j | j = 1, ..., m\}$, which are weights for selection losses over $m$ samples. The suitabilities for experts over the j-th sample, i.e. $\{S_{j,k} | k = 1, ..., n\}$ are denoted as $S_{j,:}$. A small value of the standard deviation $Std(S_{j,:})$ indicates experts have similar suitabilities on the j-th sample, thus $v_j$ gets smaller. Therefore SRM determines the value of $v_j$ by:

$$v_j = \frac{Std(S_{j,:})}{\sum_{i} Std(S_{i,:})}. \tag{7}$$

### 3.5 TRAINING METHOD

The training framework of CoE is shown in Fig 1, which consists of two stages. During the first stage, only feature extractor and task predictor of delegator are trained to minimize $\mathcal{L}_P$, cross-entropy loss for the rough prediction of delegator. During the second stage, we fix these two modules, jointly optimize the expert selector and experts with $\mathcal{L}_{Total}$:

$$\mathcal{L}_{Total} = \eta * \mathcal{L}_S + \mathcal{L}_T, \tag{8}$$

here, the hyperparameter $\eta$ is set as 0.8.

$\mathcal{L}_S$ is used to optimize the expert selector. Based on the selection label $L_{j,:}$, we can get the cross-entropy loss $\mathcal{L}_{CE}^j$ for the j-th sample. Then $\mathcal{L}_S$ is determined by the weighted sum of $\{\mathcal{L}_{CE}^j | j = 1, \dots, m\}$ with weights $\{v_j | j = 1, \dots, m\}$ output by SRM:

$$\mathcal{L}_S = \sum_j v_j * \mathcal{L}_{CE}^j. \tag{9}$$

$\mathcal{L}_T$ is used to optimize the experts. Based on the class labels of m samples, we can get $m \times n$ cross-entropy losses $\{\mathcal{L}_T^{j,k} | j = 1, \ldots, m; k = 1, \ldots, n\}$, where $\mathcal{L}_T^{j,k}$ is the cross-entropy loss for the $k$-th expert on the $j$-th sample. Then $\mathcal{L}_T$ is obtained by the weighted sum of $\mathcal{L}_T^{j,k}$ with weights $W_{j,k}$ output by WGM:

$$\mathcal{L}_T = \sum_{j,k} W_{j,k} * \mathcal{L}_T^{j,k}. \tag{10}$$

Each of our experiments includes either four or sixteen experts in this paper. When using four experts, the training method is same as described. However, we will meet slow-convergence problem if the number of experts is sixteen. To alleviate this problem, we propose a strategy which is described in Appendix A.2.

## 4 EXPERIMENTS

We conduct the main experiments on ImageNet classification task. After comparing with some popular methods in terms of computation and memory cost, we verify the superiority of CoE over some other existing model collaboration methods. Moreover, we try to generalize CoE to the translation task and re-evaluate CoE using Reassessed Labels (ReaL) (Beyer et al., 2020). Finally, we try to analyze the reasonability of learned expert selection patterns. Elaborated ablation studies are illustrated in Appendix B.4. Statistics on referenced baselines in section 4.2.1&4.2.2 are directly cited from original papers, others are implemented with the following setting unless otherwise stated.

### 4.1 IMPLEMENTATION DETAILS

We conduct experiments with two settings: **CoE-Small** and **CoE-Large**. For CoE-Small, we take TinyNet-E (Han et al., 2020b) with 24M FLOPs as the feature extractor of delegator by removing the last fully connected layer. We use OFA-110 (Cai et al., 2020) with 110M FLOPs as the expert. For the CoE-Large, MobileNetV3-Small (Howard et al., 2019) with 56M FLOPs is adopted to construct the delegator by analogy. We use OFA-230 as the experts. We have also tried to introduce CondConv (Zhang et al., 2020b) and PWLU activation fuction (Zhou et al., 2021) to achieve the extreme performance. To combine with CondConv, we replace the convolutions within each inverted residual block of the experts with CondConv ($expert\_num = 4$). To take advantage of PWLU, we replace all activation layers except those that have tiny input feature maps as illustrated in Zhou et al. (2021). Models are trained using SGD optimizer with 0.9 momentum. We use a mini-batch size of 4096, and a weight decay of 0.00002. Cosine learning rate decay is adopted and the number of training iterations is 313000. We use fixed auto-augment (Cubuk et al., 2019) as well. Inspired by BasisNet, we use knowledge distillation with EfficientNet-B2 (Tan & Le, 2019; Xie et al., 2020) as the teacher. The learning rate is 0.8/1.6 for CoE-Small/Large and dropout rate is 0.2. The stochastic depth (Huang et al., 2016) is used except for TinyNet-E with a survival probability of 0.8.

### 4.2 RESULTS AND ANALYSIS

#### 4.2.1 ACCURACY AND COMPUTATION COST

Accuracy curves for CoE in Fig.4a are drawn by varying the threshold of early termination (section 3.1). We pick out a point from each curve to compare with some efficient networks in Table 1. Our method achieves 78.2% and 80.7% top-1 accuracy with 16 experts and averaged FLOPs/Instance as 100M and 194M respectively. Compared with OFA, CoE reduces the FLOPs from 230M to 100M and from 595M to 194M, with better top-1 accuracy. Compared with EfficientNet-B1 with noisy student training, CoE also reduces the FLOPs by 3.6x while improving the accuracy by 0.5%. Though dynamic networks like GFNet, CondConv and BasisNet are more efficient than traditional networks, CoE still has significantly higher accuracy with smaller FLOPs. Compared with these approaches, CoE improves the accuracy by 2.2/2.4/0.7% respectively. When combined with CondConv, we can achieve 79.2% and 81.5% top-1 accuracy with only 102M and 214M FLOPs respectively, which indicates CoE is complementary to dynamic networks like CondConv. On the contrary, as CondConv and BasisNet share similar essence, namely using a group of basis to dynamically synthesize the input-dependent convolution kernel, the combination of them only arouses little collaborative benefit with the top-1 accuracy of only 80.5%. More surprisingly, we achieve the accuracy of 80.0% with only 100M FLOPs for the first time by further making use of PWLU.

Table 1: CoE performance on ImageNet. "CC" and "KD" indicates conditional computation approach and knowledge distillation strategy.

| Method | CC | KD | FLOPs | TOP-1 Acc |
|---|---|---|---|---|
| WeightNet (Ma et al., 2020) | √ | | 141M | 72.4% |
| DS-MBNet-M[†‡](Li et al., 2021a) | √ | √ | 319M | 72.8% |
| GhostNet 1.0x (Han et al., 2020a) | | | 141M | 73.9% |
| MobileNetV3-Large (Howard et al., 2019) | | | 219M | 75.2% |
| OFA-230 (Howard et al., 2019) | | √ | 230M | 76.9% |
| TinyNet-A (Han et al., 2020b) | | | 339M | 77.7% |
| CondConv-EfficientNet-B0 (Yang et al., 2019) | √ | | 413M | 78.3% |
| GFNet (Wang et al., 2020) | √ | | 400M | 78.5% |
| CoE-Small | √ | √ | 100M | 78.2% |
| CoE-Small + CondConv | √ | √ | 102M | 79.2% |
| CoE-Small + CondConv + PWLU | √ | √ | 100M | 80.0% |
| BasisNet (Zhang et al., 2021) | √ | √ | 198M | 80.0% |
| OFA-595 (Howard et al., 2019) | | √ | 595M | 80.0% |
| EfficientNet-B2 (Tan & Le, 2019) | | | 1.0B | 80.1% |
| EfficientNet-B1(Noisy Student) (Xie et al., 2020) | | √ | 700M | 80.2% |
| BasisNet (Zhang et al., 2021) | √ | √ | 290M | 80.3% |
| FBNetV3-C (Dai et al., 2020) | | √ | 557M | 80.5% |
| BasisNet + CondConv (Yang et al., 2019) | √ | √ | 308M | 80.5% |
| CoE-Large | √ | √ | 194M | 80.7% |
| CoE-Large + CondConv | √ | √ | 214M | 81.5% |

### 4.2.2 INFERENCE SPEED AND MEMORY COST

Compared with conditional computation methods (Yang et al., 2019; Zhang et al., 2021), CoE is more hardware friendly. To verify the advantage, we also analyze the inference latency on hardware. The experiments are conducted on CPU platform (Intel(R) Xeon(R) CPU E5-2699 v4 @ 2.20GHz) with PyTorch version as 1.8.0. We report the averaged latency on the ImageNet validation set in Table 2. We notice the discrepancy between FLOPs and real speed. For example, OFA-230 has 1.6x FLOPs compared with GhostNet 1.0x, but the speed is 1.2x faster. Moreover, this discrepancy can be enlarged by CondConv. CondConv-EfficientNet-B0 has similar FLOPs with the original EfficientNet-B0, but the speed is 1.7x slower. BasisNet synthesizes the dynamic parameters all at once, rather than the "layer by layer" fashion like CondConv, thus is more efficient. However, it still needs to load a large number of parameters for this synthesis, which brings a large MAC. This is why CoE (16 experts) can reduce 14.09% latency than BasisNet when the mini-batch size is one. Finally yet importantly, BasisNet and CondConv do not support batch processing, while CoE (16 experts) can take advantage of it (seciton 3.1) to further achieve a 3.1/6.1x speedup compared with them. We analyze the memory cost from two perspectives: the number of parameters and MAC. As can be seen from Table 2, the accuracy of CoE-Large (4 experts) is no worse than BasisNet and CondConv-EfficientNet-B0 when using similar parameters. Besides, the averaged MAC/Instance of CoE is much smaller than theirs. Compared with GhostNet 1.3x, the accuracy for Coe-Large (16 experts) is 5.0% higher with a smaller MAC.

Table 2: CPU latency and memory cost for different methods.

| Models | CPU Latency/Instance (ms) | | FLOPs | MAC | Params | Accuracy |
|---|---|---|---|---|---|---|
| | Batchsize=1 | Batchsize=64 | | | | |
| MobileNetV3-Small | 14.77 | 4.18 | 56M | 2.5M | 2.5M | 67.4% |
| GhostNet 1.0x | 39.91 | 16.50 | 141M | 5.2M | 5.2M | 73.9% |
| TinyNet-B | 34.58 | 19.44 | 202M | 3.7M | 3.7M | 75.0% |
| MobileNetV3-Large | 31.55 | 18.43 | 219M | 5.4M | 5.4M | 75.2% |
| GhostNet 1.3x | 43.94 | 29.70 | 226M | 7.3M | 7.3M | 75.7% |
| OFA-230 | 33.52 | 15.21 | 230M | 5.8M | 5.8M | 76.9% |
| EfficientNet-B0 | 49.12 | 35.21 | 391M | 5.3M | 5.3M | 77.2% |
| TinyNet-A | 45.76 | 23.71 | 339M | 5.1M | 5.1M | 77.7% |
| CondConv-EfficientNet-B0 | 81.81 | - | 413M | 24.0M | 24.0M | 78.3% |
| BasisNet | 40.61 | - | 198M | 24.9M | 24.9M | 80.0% |
| CoE-Large (4 experts) | 38.67 | 15.02 | 220M | 6.6M | 25.7M | 79.9% |
| CoE-Large (16 experts) | 34.89 | 13.30 | 194M | 6.0M | 95.3M | 80.7% |

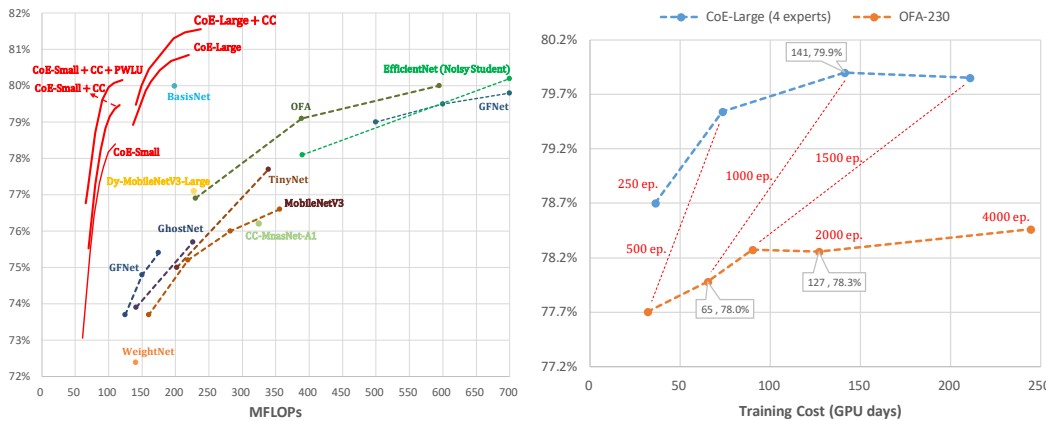

(a) Comparison with some popular networks on ImageNet. "CC" and "PWLU" means CondConv and PWLU activation function.

(b) Accuracy v.s. training cost for OFA-230 and CoE-Large (with 4 OFA-230 experts).

Figure 4: Accuracy v.s. FLOPs and training cost on ImageNet.

### 4.2.3 ANALYSIS OF THE TRAINING COST

To achieve superior performance with less inference FLOPs and latency, CoE may consume more training time. For example, the applying of CoE (4 experts) on OFA-230 improves the accuracy from 78.0% to 79.9%, but at the expense of a 2.2x training cost. To verify whether the improvement still exists with similar training cost, we get a series of accuracies by varying the number of training epochs as shown in Fig.4b, where "xx ep." means the number of training epochs is "xx" and 32 Tesla-V100-PCIe-16GB GPUs are used for training. It is seen that CoE can boost the performance (from 78.3% to 79.9%) even if the training cost is controlled to be similiar (127 vs 141 GPU days).

## 4.3 COMPARISON WITH OTHER MODEL COLLABORATION APPROACHES

### 4.3.1 COMPARISON WITH REGULAR MODEL ENSEMBLE

We train four OFA-230 models with different initialization seeds. The results are shown in Table 4. The random seed only causes minor variety in accuracy. However, an improvement of 1.6% is still achieved via ensemble. It is because these models fall into different local minima, yielding the diversity of output. Compared with the ensemble, CoE achieves 0.3/1.3% higher accuracies. It indicates CoE can realize more potential of each individual model. Besides, CoE keeps the computation cost constant, while model ensemble increases the computation cost by four times.

### 4.3.2 COMPARISON WITH MODEL SELECTION METHOD

HD-CNN (Yan et al., 2015) and HydraNets (Mullapudi et al., 2018) select branches based on the category. MoE (Shazeer et al., 2017) and Switch Transformer (Fedus et al., 2021) enable the direct training of Router by scaling the output feature of experts with gate-values predicted from Router. Despite these methods are originally designed to conditionally select a specific layer or block, we apply them to the expert selection.

To select the expert based on category, the categories should be partitioned into n groups, where n is the number of experts. We try two schemes: random partition (RP) or clustering-based partition (CBP). Then, an expert can be selected according to the rough prediction of delegator. During the training procedure, we also reweight losses of each expert based on the assignment matrix $\boldsymbol{A}_{m \times n}$ with Eq.5 and 6. $\boldsymbol{A}_{m \times n}$ is obtained directly based on the rough prediction. We can enable the direct training of expert selector as well. Directly trained selector (DTS) is obtained by scaling the loss of each expert, rather than the output feature of each branch as the original paper (Shazeer et al., 2017; Fedus et al., 2021). While CoE optimizes the expert selector via weighted cross-entropy loss $\mathcal{L}_s$ (Eq.9). The results with 4 experts are shown in Table 3. It can be seen that CoE outperforms the compared methods, demonstrating a better collaboration pattern is learned.

## 4.4 ANALYSIS OF THE GENERALIZATION

To verify the generalizability, we conduct two extra experiments: *generalizing CoE to translation task* and using Reassessed Labels (ReaL) (Beyer et al., 2020) to *re-evaluate CoE*. We mainly introduce the first one here, another one are discussed in Appendix B.1.

Table 3: Comparison with other model selection methods. "RP","CBP" and "DTS" means "Random Partition","Clustering-Based Partition" and "Directly Trained Selector" respectively.

| Method | | FLOPs | Top-1 Acc. |
|---|---|---|---|
| Category-Based | RP | 220M | 78.3% |
| | CBP | 220M | 77.5% |
| Selector-Based | DTS | 220M | 78.7% |
| | CoE | 220M | 79.9% |

Table 4: Comparison with model ensembe.

| Method | | FLOPs | Acc. |
|---|---|---|---|
| OFA-230 | Seed1 | 230M | 78.1% |
| | Seed2 | 230M | 78.0% |
| | Seed3 | 230M | 78.1% |
| | Seed4 | 230M | 78.0% |
| | Ensemble | 920M | 79.6% |
| CoE-Large | 4 Experts | 220M | 79.9% |
| | 16 Experts | 220M | 80.9% |

To generalize CoE to translation task, we build a CoE-Transformer model based on Transformer (base model) (Vaswani et al., 2017). Considering the decoding procedure is much more time-consuming than encoding because target tokens are generated one by one, only the decoder of CoE-Transformer is conditionally selected. CoE-Transformer has four decoders, given a sentence, one decoder will be selected to decode the features output by encoder. To select the decoder, an extra constant token is added at the beginning of each sentence, the feature of this token output by encoder is input to the Expert Selector (Fig.2) for expert (decoder) selection. During training, the TCP of a sentence is obtained by averaging the TCPs of each token.

Following (Vaswani et al., 2017; Ott et al., 2019), CoE-Transformer is trained on the standard WMT 2014 English-German dataset. As mentioned, an extra token will be added into this vocabulary. We adopt the same training and evaluating setting as (Ott et al., 2019), more details are shown in Appendix B.2. Although CoE-Transformer has a larger number of parameters compared with Transformer (base model), the MAC is nearly identical. We calculate MAC based on the number of parameters loaded given a sentence and report it on Table 5 as well. From Table 5 we can see, CoE-Transformer outperforms the regular Transformer (base model) by a large margin and achieves similar performance with Transformer (big).

Table 5: The BLEU scores on newstest2014 (English-to-German).

| Model | MAC | Parameters | BLEU |
|---|---|---|---|
| Transformer (base model) | 62.4M | 62.4M | 28.1 (Ott et al., 2019) |
| Transformer (big) | 213.0M | 213.0M | 29.3 (Ott et al., 2019) |
| CoE-Transformer | 62.5M | 138.2M | 29.4 |

## 4.5 ANALYSIS OF THE LEARNED EXPERT SELECTION PATTERNS

We also conduct experiments to analyze the expert selection patterns of CoE and find them quite reasonable. When experts have different architectures, the delegator learns to assign easy samples to smaller experts and complex samples to heavier experts. When experts share the same architecture, delegator learns the expert selection patterns automatically, it can be based on any property (e.g. whether humans are contained), rather than limited to the category. The details are shown in Appendix B.3.

## 5 CONCLUSION

We propose a CoE framework to pool together the expertise of multiple networks towards a common aim. Experiments in this paper demonstrate the superiority of CoE on both accuracy and real speed. We also analyze the collaboration patterns and find them has interpretability. In the future, CoE will be extended to the trillion parameters level. Meanwhile, we will try to implement CoE to more tasks and verify its compatibility with quantification and other technologies. Besides, CoE can be conducted to solve the problems of lifelong learning by updating experts.

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

## A  EXTRA DETAILS FOR METHOD

### A.1  INTRODUCTION OF VOGEL APPROXIMATION METHOD (VAM)

In weight generation module (WGM) and label generation module (LGM), we need to solve the balanced transportation problem (BTP, (Shore, 1970)) via Vogel approximation method (VAM, (Shore, 1970)). We will introduce it in this section with the number of samples and experts as $m$ and $n$ respectively.

The BTP involved in WGM and LGM has $m$ supply sources, each of which is denoted as $Silo_j$ with a supply of one, as well as $n$ demand sources, each of which is denoted as $Mill_k$ with a demand of $\frac{m}{n}$. $C_{j,k}$ is the per-unit transportation cost from $Silo_j$ to $Mill_k$. Specifically, $C_{j,k} = -P_{j,k}$ in WGM and $C_{j,k} = -S_{j,k}$ in LGM. To make it clear, we illustrate this algorithm with a toy example, where the problem is simplified as Fig.5 (a) with $m = 4$, $n = 2$. In the first step, we calculate the penalty cost $pc_{row_j}$ for each row and $pc_{col_k}$ for each column of the tableau in Fig.5 (a). Penalty cost is determined by subtracting the lowest unit cost in the row (column) from the next lowest unit cost. The penalty costs of the respective rows and columns have been marked in red color for clarity in Fig.5 (b). Since the third row has the largest penalty cost ($pc_{row_3}$ =11) and $C_{3,1}$ is the lowest unit cost of that row, $Silo_3$ is allocated to $Mill_1$, i.e. $A_{3,:} = [1, 0]$ in WGM or $L_{3,:} = [1, 0]$ in LGM. Then the corresponding row should be crossed out and the demand of $Mill_1$ should minus one, if this results in a zero demand, the first column will be crossed out as well. After adjusting penalty cost for each row and column, the tableau becomes Fig.5 (c), where the changed values are marked in orange. The described procedure will be looped until no rows remained.

Considering the calculation of $pc_{col_k}$ is much more time-consuming compared with $pc_{row_j}$ because $m \gg n$ in WGM and LGM, we modify VAM by only seeking lowest penalty cost among $\{pc_{row_1}, ..., pc_{row_m}\}$. We find this modification makes VAM more efficient while keeps the superiority of the solution. It is because the mechanics of VAM makes it meaningful to take $pc_{col_k}$ into account only when the demand of $Mill_k$ is one, which rarely happens. Thus, we adopt this modification to promote efficiency in this paper.

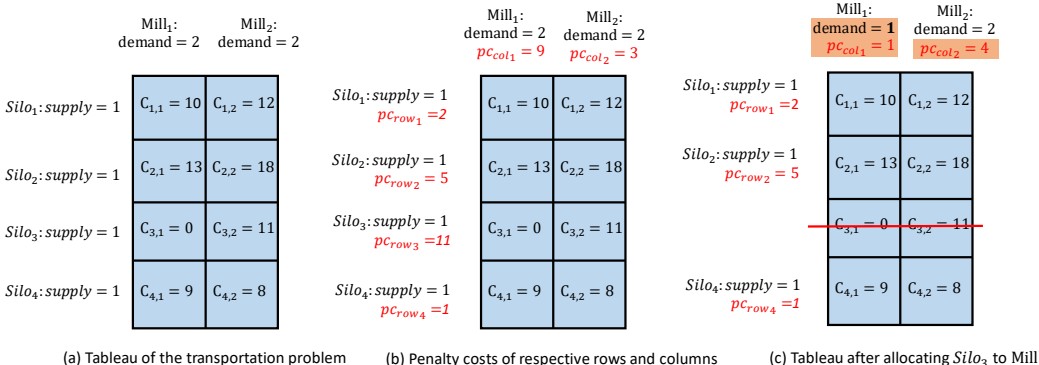

(a) Tableau of the transportation problem   (b) Penalty costs of respective rows and columns   (c) Tableau after allocating $Silo_3$ to $Mill_1$

Figure 5: The Vogel approximation method.

### A.2  A STRATEGY TO FACE THE LARGE NUMBER OF EXPERTS

Each of our experiments involves either four or sixteen experts. For the four-experts setting, the training method is the same as described. However, we will meet the slow-convergence problem when the number of experts is sixteen. It motivates us to decompose the task into four subtasks, each of which involves four experts and can be trained with the proposed training method. With this strategy, the number of samples assigned to each expert increases from $\frac{m}{16}$ to $\frac{m}{4}$. Because these samples contribute most to the optimization of the expert, the rate of convergence becomes nearly four times faster.

To fulfill task decomposition, we introduce a new module to delegator, named subtask selector as shown in Fig.6. The subtask selector is used to allocate the input samples into different subtasks. The expert selector outputs sixteen probabilities, which are partitioned into four groups as well. For

each subtask, only one group of probabilities is visible. The experts within each subtask and the corresponding weights of the expert selector are jointly optimized. As for the feature extractor, task predictor, and subtask selector, their weights directly derive from the delegator trained with the setting of four experts and then fixed. During this procedure, the weights of subtask selector derive from the expert selector.

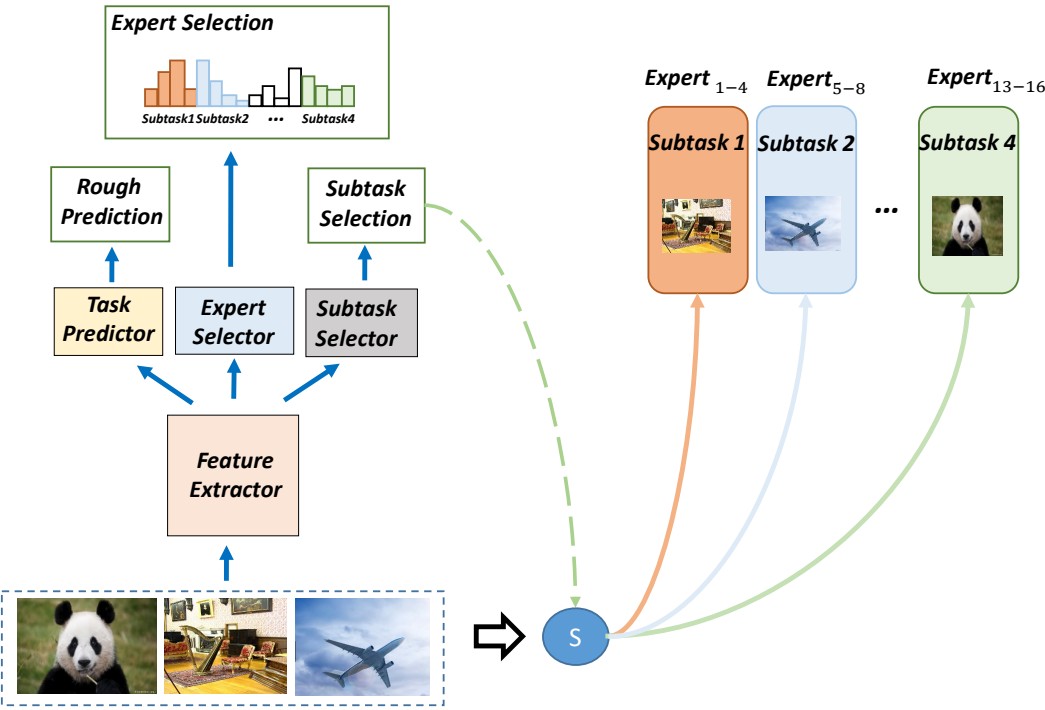

Figure 6: The modified architecture of delegator.

## B  EXTRA DETAILS FOR EXPERIMENTS

### B.1  RE-EVALUATION WITH REASSESSED LABELS

As described in paper (Beyer et al., 2020), the validation set labels have a set of deficiencies that makes the recent progress on ImageNet classification benchmark suffer from overfitting to the artifacts. To verify the generalization, we use the Reassessed Labels (ReaL) (Beyer et al., 2020) to re-evaluate our method. The results are shown in Table 6. It can be seen that our method still has a remarkable performance, achieving higher accuracy than the compared methods with significantly smaller FLOPs.

Table 6: ReaL and original top-1 accuracies. CC means CondConv.

| Method | FLOPs | ReaL Acc. | Ori. Acc. |
|---|---|---|---|
| OFA-595 (Cai et al., 2020) | 595M | 86.0% | 80.0% |
| S4L MOAM (Zhai et al., 2019) | 4B | 86.6% | 80.3% |
| ResNeXt-101 (Xie et al., 2017) | 16B | 85.2% | 79.2% |
| ResNet-152 (He et al., 2016) | 11B | 84.8% | 78.2% |
| CoE-Large | 194M | 86.5% | 80.7% |
| CoE-Large + CC | 214M | 86.9% | 81.5% |

### B.2   EXPERIMENT DETAILS FOR THE TRANSLATION TASK

Following (Vaswani et al., 2017; Ott et al., 2019), CoE-Transformer is trained on the standard WMT 2014 English-German dataset, which has a shared source-target vocabulary of about 37000 tokens. As mentioned, an extra token will be added into this vocabulary. The default training setting is identical with the one described in (Vaswani et al., 2017), except for the batch size and learning rate become larger following (Ott et al., 2019). Moreover, the parameter $\alpha$ in Eq.5 grows linearly from 0.1 to 0.4 with the training going on. We report BLEU on news2014 with a beam width of 4 and length penalty of 0.6 based on a single model obtained by averaging the last 5 checkpoints following (Vaswani et al., 2017; Ott et al., 2019).

### B.3   ANALYSIS OF THE LEARNED EXPERT SELECTION PATTERNS

#### B.3.1   EXPERT SELECTION PATTERNS WHEN EXPERTS HAVE VARIOUS ARCHITECTURES

Considering TCP can measure the complexity of a given sample if the inference model is fixed, we experiment to analyze the relationship between sample complexity and expert selection. If the experts have different computation cost, it is reasonable to assign easy samples (the ones with large TCP) to smaller experts and complex samples to heavier experts. To verify this, we take four architectures searched via OFA (Cai et al., 2020) as the experts, i.e. OFA-110, OFA-163, OFA-230 and OFA-595. The delegator is also MobileNetV3-small as described in section 4.1. We obtain the TCP value for each sample based on the delegator. We count the selection probability for each expert at different TCP values on the validation set. As shown in Fig.7, it meets our expectation that the selection probability for smaller models increases the input sample getting simpler (with the increase of TCP).

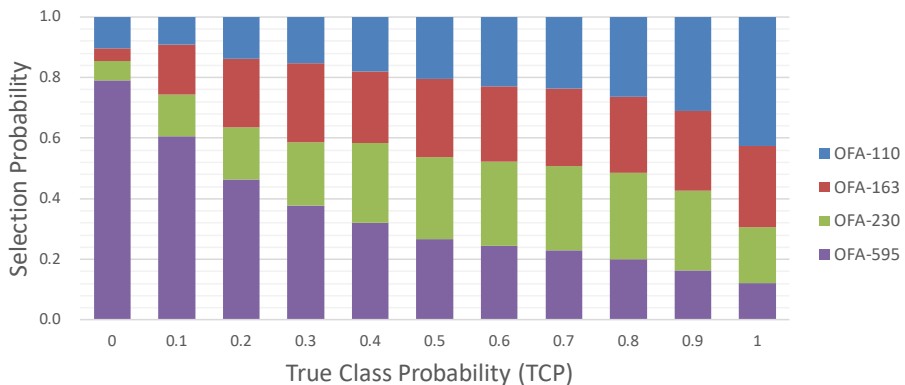

Figure 7: Selection probabilities for each expert at different TCP values.

#### B.3.2   EXPERT SELECTION PATTERNS WHEN EXPERTS SHARE THE SAME ARCHITECTURE

We have analyzed the selection patterns when the experts have different architectures, here we focus on the case that all experts share the same architecture. We adopt the CoE-Large setting with four experts.

Considering many works (Yan et al., 2015; Mullapudi et al., 2018) select branches based on the category, we firstly experiment to observe the relationship between selection patterns and rough prediction of the delegator on ImageNet validation set. Based on the predicted class of rough prediction, the validation set can be partitioned into 1000 subsets. Then we calculate the probabilities to select each expert within each subset and get 1000 probability vectors. After clustering, we plot the probability vectors on Fig.8, each column of which is a probability vector. It can be seen that samples with the same rough prediction class are assigned to different experts. Therefore, we can conclude the expert is not always selected based on category.

Besides, we further make qualitative analysis on the ImageNet validation set and find some interesting patterns. For example, we find that images predicted as "meat market" are most likely to be assigned to the fourth expert if humans are contained. We show those images in Fig.9. It can be seen, 27

images are assigned to the fourth expert, among which 22 images contain humans with a ratio of 81.5%. By contrast, among the 32 images that are assigned to the other experts, only 7 images contain humans with a ratio of 21.9%. This indicates CoE learns the expert selection patterns automatically, it can be based on properties other than the category.

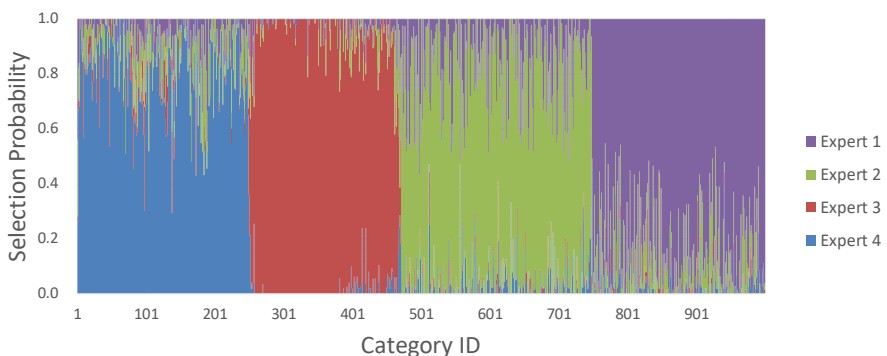

Figure 8: Selection probabilities for each expert. The horizontal axis indicates the rough prediction class. The 1000 probability vectors are clustered for better visualization.

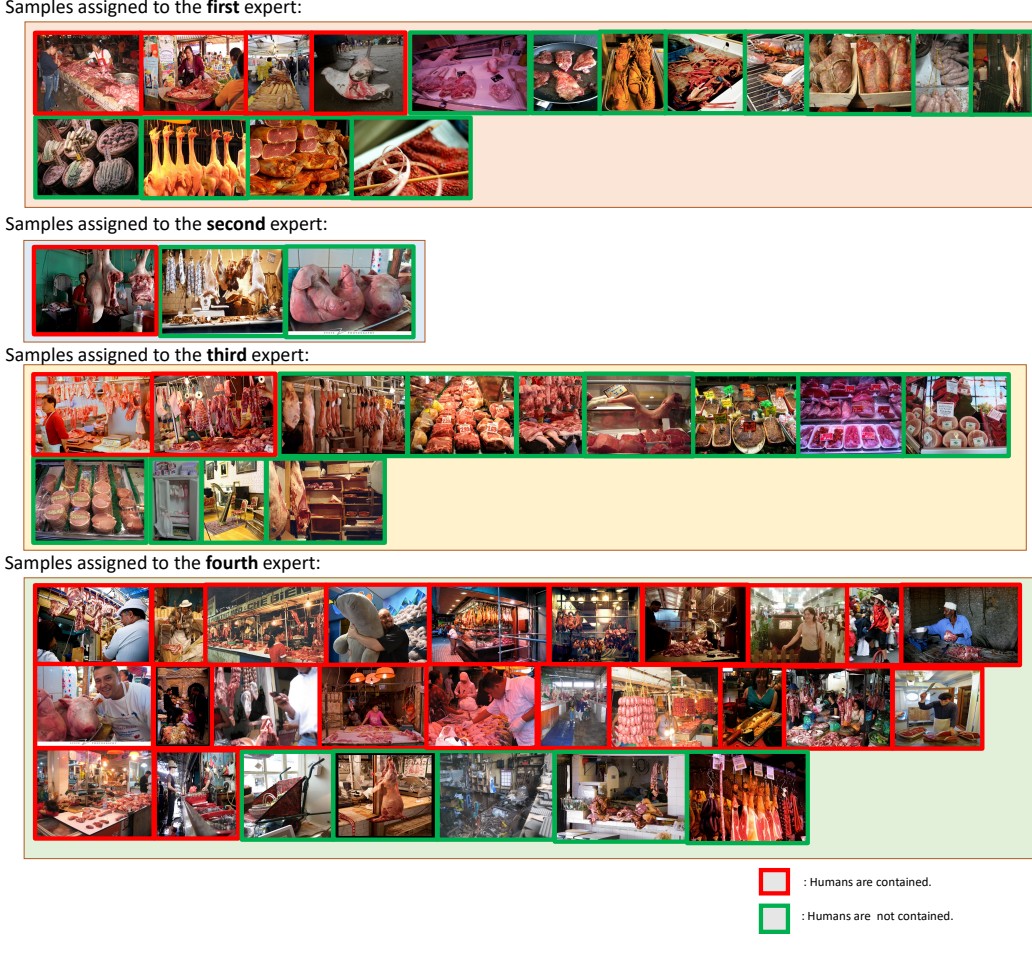

Figure 9: Images that are predicted as 'meat market' by the delegator. They are partitioned into four groups based on which expert is selected. The red border indicates humans are contained, green border indicates humans are not contained.

### B.4 ABLATION STUDIES FOR CoE

#### B.4.1 EFFECT OF EXPERT NUMBER

We analyze the number of experts in this section, including 1, 4, and 16 experts. The results are shown in Table 7. Using one expert brings little improvement compared with the original model. When increasing the number of experts, the accuracy becomes 1.9% better with four experts and 2.9% better with sixteen experts. It demonstrates CoE can make full use of multiple experts, leading to a large collaborative benefit. What's more, the accuracy also reaches 79.9% by combining CondConv with OFA-230. In this manner, CoE can further enhance the accuracy to 80.8/81.5% with 4/16 experts.

Table 7: Comparison among different number of experts. "CC" indicates CondConv.

| Method | Experts | FLOPs | Acc. |
|---|---|---|---|
| OFA-230 | - | 230M | 78.0% |
| CoE-Large | 1 | 220M | 78.0% |
| | 4 | 220M | 79.9% |
| | 16 | 220M | 80.9% |
| CC-OFA-230 | - | 242M | 79.9% |
| CoE-Large + CC | 1 | 214M | 79.9% |
| | 4 | 214M | 80.8% |
| | 16 | 214M | 81.5% |

#### B.4.2 EFFECT OF EARLY TERMINATION

The original OFA-230 has 78.0% top-1 accuracy with 230M FLOPs. We can introduce a MobileNetV3-Small to conduct early termination. By varying the threshold, we get a series of accuracies and FLOPs as shown in Fig.10. It can seen that the accuracy becomes 78.0% with 220M FLOPs. This indicates the computation cost brought by MobileNetV3-Small is eliminated via early termination strategy. Inspired by this, we expect to eliminate the computation cost brought by delegator as well. It does reduce the computation cost by 60/66M FLOPs, demonstrating the effectiveness of early termination.

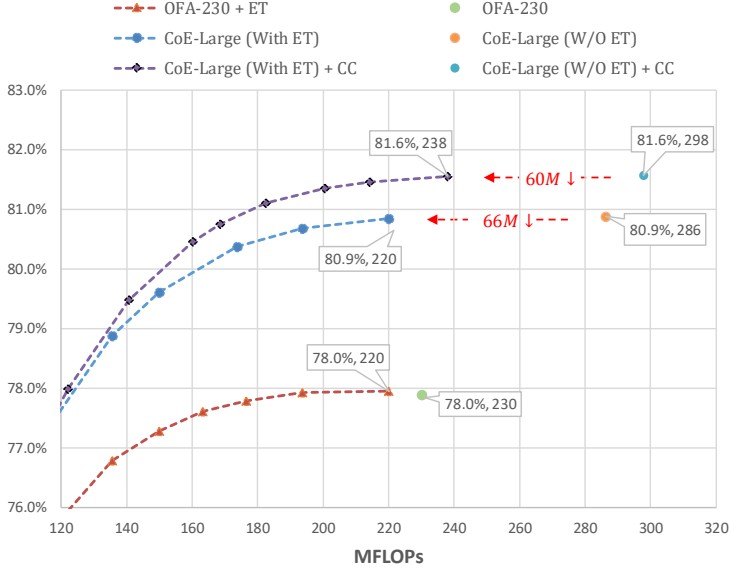

Figure 10: Accuracy v.s. FLOPs. "ET" means Early Termination and "CC" indicates CondConv.

### B.4.3 ABLATION STUDY FOR EACH COMPONENT OF CoE

CoE consists of 3 major components: WGM, LGM and SRM. Apart from directly removing each one component, we also try to alter some elements inside them. We propose several modified versions of CoE for ablation as below:

- $\text{CoE}^{WGM}$: Remove WGM from CoE. Thus, losses of experts have identical weights for each sample.
- $\text{CoE}^{WGM^\star}$: WGM partitions the training data based on expert suitability, i.e. the assignment matrix $\boldsymbol{A}_{m \times n}$ in WGM equals to the output matrix $\boldsymbol{L}_{m \times n}$ of LGM.
- $\text{CoE}^{WGM^\circ}$: Remove the "$\sum_j A_{j,k} = m/n$" constraint in Eq.4, so that $\boldsymbol{A}_{m \times n}$ neglects the *No Superiority Assumption* (NSA). Then, take $\boldsymbol{A}_{m \times n}$ as the output of WGM without the smoothing (Eq.5) and normalizing(Eq.6).
- $\text{CoE}^{WGM^\bullet}$: Remove the progressive sharpening of assignment in WGM. Specifically, using a constant 0.8 for $\alpha$ in Eq.5, instead of linearly increasing it from 0.2 to 0.8.
- $\text{CoE}^{LGM}$: Remove LGM from CoE. Thus, CoE collapse to a single expert with delegator to trigger the early termination.
- $\text{CoE}^{LGM^\star}$: Abandon the refining of suitability criterion (Eq.2) and remove the "$\sum_j L_{j,k} = m/n$"constraint in Eq.3. So that $\boldsymbol{L}_{m \times n}$ neglects the *No Superiority Assumption* (NSA).
- $\text{CoE}^{SRM}$: Remove SRM from CoE.

We conduct experiments for those CoE versions with the CoE-Large setting and 4 experts. Results are shown in Table 8, and the conclusions are listed below:

1. WGM and LGM are the most important components in CoE. The removal of WGM and LGM reduce the accuracy from 79.9% to 78.1% and 78.0%, respectively.

2. In WGM, the training data should be partitioned based on delegator. Otherwise, delegator will overfit to irregular selection labels as illustrated in section 3.3. That is why Coe-Large$^{WGM^\star}$ only achieves an accuracy of 78.4%.

3. The *No Superiority Assumption* (NSA) is important for CoE. Without this assumption, CoE-Large$^{WGM^\circ}$ and CoE-Large$^{LGM^\star}$ only reach the accuracy of 79.2% and 79.4%.

4. The progressive sharpening of assignment in WGM can also boost the performance, thus the accuracy for CoE-Large$^{WGM^\bullet}$ is 0.5% lower than CoE-Large.

5. The component SRM is also useful. It promotes delegator to select experts better, yielding a 0.4% improvement for accuracy.

Table 8: Ablations for each component of CoE.

| Method | Experts | FLOPs | Acc. |
|---|---|---|---|
| CoE-Large$^{WGM}$ | 4 | 220M | 78.1% |
| CoE-Large$^{WGM^\star}$ | 4 | 220M | 78.4% |
| CoE-Large$^{WGM^\circ}$ | 4 | 220M | 79.2% |
| CoE-Large$^{WGM^\bullet}$ | 4 | 220M | 79.4% |
| CoE-Large$^{LGM}$ | 4 | 220M | 78.0% |
| CoE-Large$^{LGM^\star}$ | 4 | 220M | 79.4% |
| CoE-Large$^{SRM}$ | 4 | 220M | 79.5% |
| CoE-Large | 4 | 220M | 79.9% |

### B.4.4 ABLATION STUDY FOR ELEMENTS OF THE TRAINING STRATEGY

Knowledge distillation (KD), auto-augment (AA) and stochastic depth (SD) are widely-used strategies to overcome the overfitting problem. We think only when the overfitting problem is solved can task accuracy reflect model capacity exactly. Because this paper is concerned with improving model

capacity with limited computation cost, we use these strategies. Nonetheless, we conduct ablations for them in this section. We adopt the CoE-Large setting and use 4 experts. Results are shown in Table 9. We find KD extremely important for CoE, it may indicate CoE is easy to be overfitted. In addition, SD decreases the accuracy of CoE. By removing SD, CoE-Large (4 experts) boosts the accuracy from 79.9% to 80.2%. Perhaps, it is because SD makes the capacity of delegator and each expert too tiny (Gontijo-Lopes et al., 2021).

Table 9: Ablation study for each component of training strategy. "KD", "AA" and "SD" denotes "knowledge distillation", "auto-augment" and "stochastic depth" respectively.

| KD | AA | SD | Experts | FLOPs | TOP-1 Acc |
|----|----|----|---------|-------|-----------|
| √ | √ | √ | 4 | 220M | **79.9**% |
| √ | √ |   | 4 | 220M | **80.2**% |
| √ |   | √ | 4 | 220M | 79.4% |
|   | √ | √ | 4 | 220M | 76.2% |
| √ |   |   | 4 | 220M | 79.7% |
|   | √ |   | 4 | 220M | **76.3**% |
|   |   | √ | 4 | 220M | 75.2% |
|   |   |   | 4 | 220M | 75.1% |

