# OpenReview forum: "Collaboration of Experts: Achieving 80% Top-1 Accuracy on ImageNet with 100M FLOPs"
_ICLR.cc/2022/Conference — ICLR 2022 Submitted_

### Official Review · Reviewer_u2in · 2021-10-29

**Correctness:** 4
**Technical Novelty And Significance:** 3
**Empirical Novelty And Significance:** 3
**Recommendation:** 5
**Confidence:** 3

**Main Review:**

Strengths:

1). The propose CoE frames achieves good performance with little computation cost.

2). The design is hardware-friendly with extreme low-latency.

3). Promoting within dataset diversity in the expert selection sounds novel.


Weakness:

1). Isn't Eq (1) encourages more randomized selection? What is the motivation of WGM, if it is refined by SRM?

2). 80.0% top-1 accuracy on ImageNet is not hard. Does the design principle can scale well across different (data size, model parameters, FLOPs)?

3). I don't know which of the proposed components contribute the most to the overall performance. What can be concluded from Appendix Figure 7 and Figure 8?


**Summary Of The Paper:**

This paper proposes the Collaboration of Experts (CoE) framework to both eliminate the need for multiple forward passes and keep hardware-friendly. A training method including weight generation module (WGM), label generation module (LGM) and selection reweighting module (SRM) is proposed to improve the expert selection. Results on ImageNet shows its hardware-friendly performance.

**Summary Of The Review:**

Good results, but lack of ablation study of each proposed components.

---

> ### Author Response · Authors · 2021-11-22
> **Reply to Reviewer u2in (1/3)**
>
> Thank you for your valuable comments. We have tried our best to revise our paper based on the reviews.
>
> To make the components WGM, LGM and SRM clearer, apart from improving the clarity, we also try to provide some in-depth analysis about ''why they work?'' and ''why they are designed like this?''. We introduce ''what they aim to do'' in ''abstract''. Then, the third paragraph of ''introduction'' briefly illustrates ''what they do'' along with ''why they work''. In ''method'' (section 3.2, 3.3 and 3.4), we try to improve the clarity of our statements about their details. In the first three paragraphs of section 3.3, we analyze why delegator and experts learn to collaborate, in addition to the collapse caused by a different data partition manner (the one LGM adopts) in WGM with a demo Fig.3. Meanwhile, elaborated ablation studies are also added for each of these components appendix B.4.3 (the ablation studies for elements of training are also added in B.4.4). We try to answer your questions as below.
>
>
> **Q1:** Isn't Eq (1) (Eq.4 of the revision) encouraging more randomized selection?
>
> **A1:**  We are sorry for the unclarity. This equation encourages the dataset to be partitioned by maximizing the sum of probabilities $\displaystyle P_{m\times n}$ output by delegator on m samples, random selection is not encouraged here. This equation generates an assignment matrix $\displaystyle A_{m\times n}$, which indicates the partition of the dataset. $\displaystyle A_{m\times n}$ has one-hot row vectors, $A_{j,k} = 1$ means the j-th sample $x_j$ is assigned to the k-th expert, thus WGM enlarges the loss weight for $\text{Expert}_k$ on $x_j$.
>
> By the way, there may arise the question of why delegator and experts learn to collaborate, and we introduce it here. LGM generates the label (selection label) to constitute the loss of delegator for expert selection (selection loss). The selection label is a one-hot vector, indicating the suitable expert for each given input. Due to the random initialization of experts, selection labels are irregular in the early training stage. Nonetheless, delegator tends to learn generalizable patterns first, since networks learn gradually more complex hypotheses during training [A]. With delegator as the bridge, WGM can partition the training data into portions based on generalizable patterns, and reweight expert losses to impel each expert to focus on one portion. This procedure makes selection labels more regular in return. Thus, delegator avoids overfitting to the irregular labels and learns to select experts properly. This procedure is visualized in Fig.3b of the revision.
>
>
> **Q2:** What is the motivation of WGM?
>
> **A2:**  To maximize the collective capacity of CoE, each expert is encouraged to focus on one unique portion of the dataset. This is achieved by WGM, which firstly partitions the dataset into portions then reweights losses of experts to impel each expert to focus on one portion.
>
> As illustrated in ''A1'', the partition in WGM is based on delegator and indicated by $\displaystyle A_{m\times n}$. Based on $\displaystyle A_{m\times n}$, WGM generates the loss weights $\displaystyle W_{m\times n}$ via smoothing and normalizing  (Eq.5\&6 of the revison).
>
> **Q3:** Is WGM refined by SRM?
>
> **A3:** We are sorry for the unclarity. WGM reweights losses of **experts** to impel each expert to focus on one portion of the dataset, while SRM reweights losses of **delegator** to enforce delegator to focus on samples whose recognition results are sensitive to expert selection. Therefore, WGM is not refined by SRM.
>
> By the way, there may arise the question of why SRM is useful. We try to illustrate it here. Expert suitability can be measured with standardized TCP (Eq.2 of the revision). If experts have similar suitabilities for a given sample, the expert selection will have little influence on the final performance of CoE. As the capacity of delegator is limited, it should pay less attention to those samples. This is achieved by SRM, which reweights losses of delegator based on a statistic that reflects suitability similarity (Eq.7 of the revision). The suitabilities for experts over the j-th sample, i.e. {$S_{j,k}|k=1,...,n$} are denoted as $S_{j,:}$. A small value of the standard deviation $\text{Std}(S_{j,:})$ indicates experts have similar suitabilities on the j-th sample, thus the weight for loss of delegator gets smaller. Moreover, the added ablation study in Appendix B.4.3 shows SRM can improve the performance of CoE.

---

> > ### Author Response · Authors · 2021-11-22
> > **eply to Reviewer u2in (2/3)**
> >
> > **Q4**: 80.0\% top-1 accuracy on ImageNet is not hard. Can the designed principle scale well across different (data size, model parameters, FLOPs)?
> >
> > **A4**:  Due to the resource constraint, we can only afford to verify CoE in the mobile setting. But we believe in the potential of CoE for scaling to big models with trillion parameters. If there is a chance in the future, we look forward to extending CoE to large models. By the way, we conduct experiments with two settings: CoE-Small and CoE-Large. They vary in parameters and FLOPs, the results (Table 1\& Fig.4a of the revision) demonstrate CoE can scale well across different FLOPs and parameters in mobile setting.
> >
> >
> > **Q5**: Which of the proposed components contribute the most to the overall performance. This paper lacks the ablation study of each proposed component.
> >
> > **A5**: Thanks for your comment which improves the completeness of this paper. We have added the ablation study in Appendix B.4.3 of the revision. CoE consists of 3 major components: WGM, LGM and SRM. Apart from directly removing each one component, we also try to alter some elements inside them. We propose several modified versions of CoE for ablation as below:
> > * **CoE**$^{WGM}$ : Remove WGM from CoE. Thus, losses of experts have identical weights for each sample.
> > * **CoE**$^{WGM^\star}$: WGM partitions the training data based on expert suitability, i.e. the assignment matrix $\displaystyle A_{m\times n}$ in WGM equals the output matrix $\displaystyle L_{m\times n}$ of LGM.
> >
> > * **CoE**$^{WGM^\circ}$: Remove the ''$\sum\_{j}A\_{j,k} =m/n$'' constraint in Eq.4, so that $\displaystyle A_{m\times n}$ neglects the **No Superiority Assumption (NSA)**. Then, take $A_{m\times n}$ as the output of WGM without the smoothing and normalizing(Eq.5&6 of the revision).
> >
> > * **CoE**$^{WGM^\bullet}$: Remove the progressive sharpening of assignment in WGM. Specifically, using a constant 0.8 for $\alpha$ in Eq.5, instead of linearly increasing it from 0.2 to 0.8.
> >
> > * **CoE**$^{LGM}$: Remove LGM from CoE. Thus, CoE collapse to a single expert with delegator to trigger the early termination.
> >
> > * **CoE**$^{LGM^\star}$: Abandon the refining of suitability metric (Eq.2) and remove the ''$\sum\_{j}L\_{j,k} =m/n$''constraint in Eq.3. So that $\displaystyle L_{m\times n}$ neglects the **No Superiority Assumption（NSA)**.
> >
> > * **CoE**$^{SRM}$: Remove SRM from CoE.
> >
> > | Method| Experts| FLOPs|TOP-1 Acc.|
> > | :----:|    :----:   |          :----: |          :----: |
> > | **CoE-Large**$^{WGM}$| 4|220M   |78.1\%|
> > | **CoE-Large**$^{WGM^\star}$| 4|220M   |78.4\%|
> > | **CoE-Large**$^{WGM^\circ}$| 4|220M   |79.2\%|
> > | **CoE-Large**$^{WGM^\bullet}$|   4 |220M   |79.4\%|
> > | **CoE-Large**$^{LGM}$| 4   |220M   |78.0\%|
> > | **CoE-Large**$^{LGM^\star}$| 4   |220M   |79.4\%|
> > | **CoE-Large**$^{SRM}$| 4   |220M   |79.5\%|
> > | **CoE-Large**|   4  |220M   |79.9\%|
> >
> > We conduct experiments for those CoE versions with the **CoE-Large** setting and **4 experts**. Results are shown in the table above, and the conclusions are listed below:
> >
> > * WGM and LGM are the most important components in CoE. The removal of WGM and LGM reduce the accuracy from 79.9\% to 78.1\% and 78.0\%, respectively.
> >
> > *  In WGM, the training data should be partitioned based on delegator. Otherwise, delegator will overfit to irregular selection labels as illustrated in section 3.3. That is why **Coe-Large**$^{WGM^\star}$ only achieves an accuracy of 78.4\%.
> >
> > * The **No Superiority Assumption (NSA)** is importart for CoE. Without this assumption, **CoE-Large**$^{WGM^\circ}$ and **CoE-Large**$^{LGM^\star}$ only reach the accuracy of 79.2\% and 79.4\%.
> >
> > * The progressive sharpening of assignment in WGM can also boost the performance, thus the accuracy for **CoE-Large**$^{WGM^\bullet}$ is 0.5\% lower than **CoE-Large**.
> >
> > * The component SRM is also useful. It promotes delegator to select experts better, yielding a 0.4\% improvement for accuracy.

---

> > > ### Author Response · Authors · 2021-11-22
> > > **Reply to Reviewer u2in (3/3)**
> > >
> > > **Q6**: What can be concluded from Appendix Figure 7?
> > >
> > > **A6**: In Figure 7, the experts have different FLOPs. It is reasonable to assign easy samples to smaller experts and complex samples to heavier experts. We experiment to verify whether delegator learns this expert selection pattern. It meets our expectation that the selection probability for smaller models increases with the input sample getting simpler (a large TCP means the input is simple when the inference model is fixed). Thus we conclude that delegator can learn reasonable expert selection patterns.
> > >
> > > **Q7**: What can be concluded from Appendix Figure 8?
> > >
> > > **A7**:  Considering many works [B, C] select branches (experts) based on category, we experiment to verify whether CoE shares the same essence and show the result in Figure 8. It shows samples with the same predicted class are assigned to different experts thus we conclude that CoE learns the model selection patterns automatically, it can be based on any property, rather than limited to the category.
> > >
> > > By the way, we introduce how is Figure 8 obtained here. Based on the predicted class derived from rough prediction, the ImageNet validation set can be partitioned into 1000 subsets. Then we calculate the probabilities to select each expert within each subset and get 1000 probability vectors. After clustering, they are plotted in Fig.8, thus each column of Fig.8 is a probability vector.
> > >
> > >
> > > ---
> > >
> > > [A] Devansh Arpit, et al. A closer look at memorization in deep networks. In ICML, 2017.
> > >
> > > [B] HD-CNN: hierarchical deep convolutional neural networks for large-scale visual recognition. ICLR 2015.
> > >
> > > [C] HydraNets: Specialized dynamic architectures for efficient inference. CVPR 2018.
> > >
> > > ---

---

### Official Review · Reviewer_u2o7 · 2021-11-01

**Correctness:** 3
**Technical Novelty And Significance:** 2
**Empirical Novelty And Significance:** 3
**Recommendation:** 5
**Confidence:** 3

**Main Review:**

++ The paper proposes a novel approach to partitioning the entire dataset to individual experts while maintaining diversity.

++ The expert selection is done at the model level, so that the advantage of state-of-the-art efficient networks (OFANet, MobileNetV3) can be taken for small FLOPs and low CPU latency.

++ The proposed CoE is also applied to translation task with BERT and shows generalization ability.

-- Method: The module names are unclear. It makes me hard to understand the role until I read the main text. Furthermore, even if I go through the writing, I am still unsure what is the true difference between selection matrix $ A_{m\times n} $  and the selection label $ S_{m\times n} $. Are they the same thing but optimized separately by two different constraints (Eq.1 and 6)? Also, the authors didn't clearly specify how the inference works.

-- Why does the "Collaboration" means in the framework "Collaboration of Experts"? The selection is one-hot, so there is actually no collaboration between experts?

-- Unclear comparison.
 (1) Table 1 shows CoE-Small/Large with both Conditional Computation (CC) and Knowledge Distillation (KD). I am curious the result of CoE-Small/Large without KD.
  (2) CoE-Large use OFA-230 as the expert network, so how could CoE-Large achieve smaller FLOPs compared with OFA-230? Is it also including the early termination?

-- Style of writing: A pair of parentheses over all references is preferred so that the the references will not be confused with the main text. I think "\citep" can do this.


**Summary Of The Paper:**

The paper proposes an efficient mechanism for  image classification. It has a light-weight delegator to yield coarse prediction, where early termination is possible. Once the coarse prediction is not confident enough, it actuates an down-stream expert with higher computational power for refined prediction. Since the expert selection is done across models, the method can take advantages of heavily-tuned efficient neural networks. The so-called CoE framework achieves very competitive results on the ImageNet dataset with low FLOPs and low CPU latency.

**Summary Of The Review:**

The paper combines the recent advances of efficient (low FLOPs and low latency) networks and some other techniques such as knowledge distillation, conditional conv (CondConv), and highly-engineering activation function. What is the true novelty is somewhat unclear to me. Also, the methods are not clearly written and how different modules work is confusing.

---

> ### Author Response · Authors · 2021-11-22
> **Reply to Reviewer u2o7 (1/2)**
>
> We would first like to thank you for your valuable comment and suggestion. We have tried our best to revise our paper based on the reviews.
>
> To make the components of CoE (i.e. WGM, LGM and SRM) easy to understand, apart from improving the clarity, we also try to provide some in-depth analysis about ''why they work?'' and ''why they are designed like this?''. We introduce ''what they aim to do'' in ''abstract''. Then, the third paragraph of ''introduction'' briefly illustrates ''what they do'' along with ''why they work''. In ''method'' (section 3.2, 3.3 and 3.4), we try to improve the clarity of our statements about their details. In the first three paragraphs of section 3.3, we analyze why delegator and experts learn to collaborate, in addition to the collapse caused by a different data partition manner (the one LGM adopts) in WGM with a demo Fig.3.
>
> Moreover, we have tried to improve the clarity by revising the English and the style of writing. For example, we have modified the references according to your valuable suggestion. “\citep” is used and “\citecolor” are set as violet to avoid confusion. We have also added more descriptions about the inference procedure. Thanks for your suggestion, we conduct the ablation studies for training strategies (including KD) which are analyzed in Appendix B.4.4. We try to answer your questions as below.
>
> **Q1:** What is the true difference between assignment matrix $\displaystyle A_{m\times n}$ and selection label $\displaystyle L_{m \times n}$.
>
> **A1:** The core difference between assignment matrix $\displaystyle A_{m\times n}$ and selection labels $\displaystyle L_{m \times n}$ is what objective function they are optimized by. $\displaystyle L_{m \times n}$ is optimized by maximizing the sum of standardized TCP ($S_{j,k}$ in Eq.2 of the revision) of selected experts on m samples, while $\displaystyle A_{m\times n}$ aims to maximize the sum of probabilities output by delegator.
>
> If the assignment matrix $\displaystyle A_{m\times n}$ is optimized by standardized TCP as well, $\displaystyle A_{m \times n}$ will be equal to $\displaystyle L_{m \times n}$. That means the dataset is partitioned into portions based on $\displaystyle L_{m \times n}$, then WGM reweights the losses of experts to impels each expert to focus on one portion. As illustrated in section 3.3 of the revision, delegator will get caught in an overfitting problem. Assuming $Expert_k$ is suitable on a sample $x_j$ , thus $A_{j,k}=L_{j,k}=1$. Due to $A_{j,k}=1$, the loss weight for $\text{Expert}_k$ gets larger than other experts on $x_j$, making $\text{Expert}_k$ more suitable in return. Therefore selection labels cannot be updated. Moreover, selection labels are irregular in the early training stage because of the random initialization, thus selection labels will remain irregular consistently. With the training going on, delegator will overfit those irregular labels, yielding poor generalization as shown in Fig.3a. As a result, CoE performs poorly because delegator can hardly select a suitable expert during validation. This is verified in Appendix B.4.3 (CoE-Large$^{WGM^\star}$).
>
> Since networks learn gradually more complex hypotheses during training [A], delegator tends to learn generalizable patterns first. Therefore, the partition can be based on generalizable patterns with delegator as the bridge. In this way, selection labels get more regular in return thanks to the reweighting of expert losses in WGM. As shown in Fig.3b, delegator avoids overfitting to the irregular labels, hence generalizing well to the validation set.
>
> **Q2:** How the inference works.
>
> **A2:** We have revised our paper to improve the clarity of the inference procedure of CoE. A new section 3.1 is added to specify how the inference works. Given a sample, the inference procedure can be summarized as below:
>
> ---
>
> **Input:** a threshold $\tau \in [0, 1]$, a sample $x$, the number of classes $C$;
>
> **Output:** the predicted class label $Y$;
>
> ---
>
> **Procedure:**
>
> - **Step1** Get the rough prediction $Pred^{rough}$ and determine the selected expert $expert_k$ for the sample $x$ , where $Pred^{rough}_c$ denotes the probability of the $c$-th class.
>
> - **Step2** Calculate the MCP value, $MCP=Max(Pred_c^{rough}|c=1,...,C)$.
>
> - **Step3** If $MCP>\tau$, $Y=Argmax(Pred^{rough})$; else goto Step4.
>
> - **Step4** Get the refined prediction $Pred^{refined}$ for the sample $x$ with $expert_k$.
>
> - **Step5** $Y=Argmax(Pred^{refined})$.
>
> By varying the threshold $\tau$, we can get a series of accuracies with averaged FLOPs/Instance from $F_D$ to ($F_D$ + $F_E$), where $F_D$ and $F_E$ are FLOPs of delegator and expert.

---

> > ### Author Response · Authors · 2021-11-22
> > **Reply to Reviewer u2o7 (2/2)**
> >
> > **Q3:**  What is the true novelty (the question in '''Summary Of The Review').
> >
> > **A3:** We are sorry for the unclarity, the novelty of this paper is summarized as below:
> >
> > * We propose a collaboration framework named Collaboration of Experts (CoE) to do tiny ML effectively. The core advantage of it is the inference efficiency. Compared with other conditional computation methods, CoE has a low memory access cost and a high degree of parallelism, which are two important factors for real latency [C].
> >
> >
> > * We present a novel optimization strategy for CoE. Without PWLU activation function and CondConv, CoE is still the first work reaching 80.7\% Top-1 accuracy on ImageNet with less than 200M FLOPs as far as we know. Moreover, the added ablations in Appendix B.4.3 demonstrate the significance of each component (WGM, LGM and SRM) as well as some elements inside them (e.g. the progressive sharpening of assignment in WGM, the No Superiority Assumption and the dataset partition basis).
> >
> >
> > **Q4:**  What does "Collaboration" mean in the framework "Collaboration of Experts"?
> >
> > **A4:** Experts are collaborated to solve the whole task. As shown in Appendix B.3, delegator learns to select experts based on the sample complexity or properties like whether humans are contained. It demonstrates each expert is responsible for a unique portion of the dataset, this is also a kind of “model collaboration” with the delegation scheme.
> >
> > **Q5:**  The result of CoE without KD.
> >
> > **A5:**  We have added the ablation study for knowledge distillation (KD), auto-augment (AA) and stochastic depth (SD) in appendix B.4.4 of the reversion. We adopt the CoE-Large setting and use 4 experts. We find KD extremely important for CoE, it may indicate CoE is easy to be overfitted. In addition, SD decreases the accuracy of CoE. By removing SD, CoE-Large (4 experts) boosts the accuracy from 79.9\% to 80.2\%. Perhaps, it is because SD makes the capacity of delegator and each expert too tiny [B]. By the way, KD is a widely-used strategy to overcome the overfitting problem. We think only when the overfitting problem is solved can task accuracy reflect model capacity exactly.  Because this paper is concerned with improving model capacity with limited computation cost, we use it in the original paper.
> >
> > | KD| AA | SD     |Experts|FLOPs|TOP-1 Acc.|
> > | :----:|    :----:   |          :----: |          :----: |          :----: |:----: |
> > | ✓| ✓| ✓   |4|220M   |**79.9\%**|
> > | ✓| ✓|    |4|220M   |**80.2\%**|
> > | ✓| | ✓   |4|220M   |79.4\%|
> > | | ✓| ✓   |4|220M   |76.2\%|
> > | ✓| |   |4 |220M   |79.7\%|
> > | | ✓| |4|220M   |**76.3\%**|
> > | | | ✓  |4 |220M   |75.2\%|
> > | | |    |4|220M   |75.1\%|
> >
> > **Q6:**  How could CoE-Large achieve smaller FLOPs compared with OFA-230?
> >
> > **A6:**   It is because CoE uses early termination during inference. As mentioned in section 3.1 of the revision, by varying the threshold of early termination, CoE-Large can get a series of accuracies with averaged FLOPs/Instance from 56 M to (56 + 230) M.
> >
> > **Q7:**  A pair of parentheses over all references is preferred so that the references will not be confused with the main text.
> >
> > **A7:**  We have revised the references of the revision. ''\citep'' is used and ''\citecolor'' is set as violet to avoid confusion.
> >
> > ---
> >
> > [A] Devansh Arpit, et al. A closer look at memorization in deep networks. In ICML, 2017.
> >
> > [B] Tradeoffs in data augmentation: An empirical study. ICLR 2021.
> >
> > [C] Shufflenet v2: Practical guidelines for efficient CNN architecture design.
> >
> > ---

---

### Official Review · Reviewer_SUb5 · 2021-11-02

**Correctness:** 4
**Technical Novelty And Significance:** 4
**Empirical Novelty And Significance:** 4
**Recommendation:** 8
**Confidence:** 5

**Main Review:**

Let me first say that I like the paper and I think it can be very valuable to the community. It is practical, it performs really well, and achieves surprising results on a very crowded, high-impact hot topic, which is impressive. That said, there are some very important issues with the explanations and even with the experimental comparisons (despite some good range of ablations). My initial score is very likely to change after rebuttal depending on the authors reply.

Here are some of my issues:

- What the WGM, LGM and SRM do is not clear. There are some verbal descriptions, which do not clarify much, and the maths is not very clear either. The WGM (btw weight generation module, the name seems non-informative at best to me, assignment generation module? assignment balancing module?) seems to do something very similar to the LGM. My understanding is that the LGM assigns an expert to each training instance, thus generating the labels that are used to train the delegator. But what is the role of the WGM? is it balancing so that there is no collapse to a single expert? I am not sure I get why the WGM is needed and what it does in practice.
I think I understand what the SRM module does, but I'm not sure either. Why is it needed? My best guess given the first paragraph of sec 3.3 is that it softens the penalty of delegator mistakes based on how well the "wrong" experts do. But then, why use an indirect metric like statistics instead of using the actual loss of the chosen expert?

- Equations: Eq. 10 has a \mathcal{L}_j,k, but that's not defined before. I assume it is L_j,k as defined in the LGM? What is \mathcal_P in the first paragraph of Sec 3.4? Not defined or used either. Is it the same as \mathcal{L}_{total}?

- Sec. 4.1 shows the FLOPs of the networks involved. However, the delegator is 24MB and the expert 110MB. How do we get to 100MB?

- The early termination (early exit?) is virtually not mentioned in the whole paper. There is no explanation regarding whether/how/when it is used, training details, etc. Reading sec. 4.2.3 I recalled it being mentioned in the intro, but at that point I had forgotten about it.

- There is no mention on the paper on weather you would release the code upon acceptance. Given that some parts of the paper/method are hard to parse, that would be very helpful. Is this expected?

There are other less critical comments:
- batching: it is mentioned that the proposed method is better at batching than other conditional convolution methods. However, it is still the case that different examples in a batch will go to different experts, thus negatively affecting batching vs. standard networks. Is this correct?
- I would like to see MobileNetV3-large Table 2
- what is the performance without KD? Can you ablate it?
- Did you ablate the elements on the optimization? I was very surprised at things like stochastic depth and autoaugment given the very tiny capacity fo the network. Are these elements really necessary? (e.g. see tradeoffs in data augmentation: an empirical study, ICLR'21).
- There are some phrases that are hard to understand. The clarify of the paper would improve revising its English (this is not a factor for scoring, just advice on how to improve impact).
- Figures are too small, they cannot be read when printed. The only way to see them properly is to zoom (a lot) into the pdf.
- There are a large number of ablations and supplementary experiments. This thoroughness is appreciated. However, it would be interesting to see an ablation on some of the components of the model - for example, how important are the SRM? and the WGM? and the progressive sharpening of the assignments? etc (I guess this one relates to the KD ablation comment above). I understand this might take resources and time to run and might not fit into a rebuttal, but overall I think it is something missing.

Now, there's plenty of positives with the paper. The performance is excellent on a very important problem (how to do tiny ML effective). The awareness of the literature is superb. There are loads of experiments (CPU latency, vit...). There is solid novelty - I think many people have thought of this way of tackling the problem, but somehow this paper explains how to make it work, so extra kudos. I also like the aggregation of other pieces to build a very solid system (CC, KD, PWLR)... our field tends to focus on one single novelty per paper to maximize clarity (and avoid confusing reviewers?), but then it is very unclear how things stack up and leaves a lot of gap that needs to be filled by the practitioner. I like that this paper bridges the gap and offers a "ready to go" model.

So, all in all, I would really like clarity improved because I think the paper really deserves it. If that happens, I'll happily improve my score.


**Summary Of The Paper:**

The paper proposes a model for classification (on ImageNet) that involves having a very small network act as a selector and early predictor, and a set of experts that specialise on a subset of the training data. The selector chooses one of them and thus keeps a constant cost irrespective of the number of experts. This is then combined with a number of existing techniques to raise the overall performance. Namely, the authors use a PWRL non-linearity instead of ReLU, they use logit-matching knowledge distillation, and conditional convolution (CondConv). The outcome is a tiny model that yields excellent performance on ImageNet, with performance envelopes superior to existing methods.

**Summary Of The Review:**

The paper has a lot of pros and cons. The cons look solvable (clarity mostly), and the pros seem solid enough. The practicality of the approach, the thoroughness of experiments and comparisons, and the excellent performance attained are all very positive. As already mentioned, a good rebuttal that shows clearer explanations will mean I raise the score.

---

> ### Author Response · Authors · 2021-11-22
> **Reply to Reviewer SUb5(1/3)**
>
> We would first like to thank you for the positive support. Your remarks are thorough and helpful. We have revised our paper based on the reviews and tried our best to improve the clarity.
>
> To make the components of CoE (i.e. WGM, LGM and SRM) easy to understand, apart from improving the clarity, we also try to provide some in-depth analysis about ''why they work?'' and ''why they are designed like this?''. We introduce ''what they aim to do'' in ''abstract''. Then, the third paragraph of ''introduction'' briefly illustrates ''what they do'' along with ''why they work''. In ''method'' (section 3.2, 3.3 and 3.4), we try to improve the clarity of our statements about their details. In the first three paragraphs of section 3.3, we analyze why delegator and experts learn to collaborate, in addition to the collapse caused by a different data partition manner (the one LGM adopts) in WGM with a demo Fig.3.
>
> We have also revised the statement in section 3.5 to describe the training losses more clearly and added more descriptions about the inference procedure. Thanks for your suggestion, we conduct the ablation studies which are analyzed in Appendix B.4.3 and B.4.4. We try to answer your questions as below.
>
> **Q1:** The difference between WGM and LGM.
>
> **A1:** LGM is used to generate the label (selection label) to constitute the loss of delegator for expert selection (selection loss). Meanwhile, WGM firstly partitions the dataset into portions then encourages each expert to focus on one portion by reweighting losses of experts. The core difference between WGM and LGM in terms of assigning experts is what they are based on. LGM assigns experts based on standardized TCP ($S\_{j,k}$ in Eq.2 of the revision), while WGM partitions the dataset based on the probabilities for expert selection output by delegator. As analyzed in section 3.3 of the revision, If WGM partitions the dataset based on standardized TCP as well, a poor generalization will be caused for delegator.
>
> The partition in WGM can be indicated by an assignment matrix $\displaystyle A\_{m\times n}$, with one-hot row vectors. $A\_{j,k} = 1$ means the j-th sample $x\_j$ is assigned to the k-th expert, thus the loss weight for $\text{Expert}\_k$ gets larger than other experts on $x\_j$. If WGM partitions the dataset based on standardized TCP as well, $\displaystyle A\_{m\times n}$ will be equal to the selection labels $\displaystyle L\_{m\times n}$ output by LGM. Assuming $Expert\_k$ is suitable on a sample $x\_j$, thus $A\_{j,k}=L\_{j,k}=1$. Due to $A\_{j,k}=1$, the loss weight for $\text{Expert}\_k$ gets larger than other experts on $x\_j$, making $\text{Expert}\_k$ more suitable in return. Therefore selection labels cannot be updated. Moreover, selection labels are irregular in the early training stage because of the random initialization, thus selection labels will remain irregular consistently. With the training going on, delegator will overfit those irregular labels, yielding poor generalization as shown in Fig.3a of the revision. As a result, CoE performs poorly because delegator can hardly select a suitable expert during validation. This is also verified in Appendix B.4.3 (CoE-Large$^{WGM^\star}$).
>
> Since networks learn gradually more complex hypotheses during training [A], delegator tends to learn generalizable patterns first. Therefore, the partition can be based on generalizable patterns with delegator as the bridge. In this way, selection labels get more regular in return thanks to the reweighting of expert losses in WGM. As shown in Fig.3b of the revision, delegator avoids overfitting to the irregular labels, hence generalizing well to the validation set.
>
> Moreover, WGM not only partitions the dataset but also generates the final loss weights for experts via smoothing and normalizing (Eq.5\&6 of the revision).
>
> **Q2:**  Why SRM is needed.
>
> **A2:** Expert suitability can be measured with standardized TCP (Eq.2 of the revision). If experts have similar suitabilities for a given sample, the expert selection will have little influence on the final performance of CoE. As the capacity of delegator is limited, it should pay less attention to those samples. This is achieved by SRM, which reweights losses of delegator based on a statistic that reflects suitability similarity (Eq.7 of the revision). The suitabilities for experts over the j-th sample, i.e. {$S\_{j,k}|k=1,...,n$} are denoted as $S\_{j,:}$. A small value of the standard deviation $\text{Std}(S\_{j,:})$ indicates experts have similar suitabilities on the j-th sample, thus the weight for loss of delegator gets smaller. Moreover, the added ablation study in Appendix B.4.3 shows SRM can improve the performance of CoE.

---

> > ### Author Response · Authors · 2021-11-22
> > **Reply to Reviewer SUb5(2/3)**
> >
> > **Q3:**  Equations in section 3.4 (section 3.5 for revision) need to be clarified.
> >
> > **A3:**  We show the definition and usage of $\mathcal{L}\_{P}$,  $\mathcal{L}\_{j,k}$ and $\mathcal{L}\_{\text{Total}}$ in Figure.1, but do not illustrate them properly in the main text of the original paper. We have added this in section 3.5 of the revision and renamed  $\mathcal{L}\_{j,k}$ as $\mathcal{L}\_T^{j,k}$ to avoid confusion.
> >
> > * $\mathcal{L}\_{P}$ : Besides expert selection, delegator also outputs a rough prediction. $\mathcal{L}\_{P}$ is the cross-entropy loss for this rough prediction. It is used to optimize the delegator.
> >
> > * $\mathcal{L}\_T^{j,k}$ : Given $m$ samples and $n$ experts, we can get $m \times n$ cross-entropy losses. Among which, $\mathcal{L}\_T^{j,k}$ is the cross-entropy loss for the $k$-th expert on the $j$-th sample and used to optimize the $k$-th expert.
> >
> > *  $\mathcal{L}\_{\text{Total}}$ : $\mathcal{L}\_{\text{Total}}$ is obtained by combining the expert selection loss $\mathcal{L}\_{\text{S}}$ of delegator and the cross-entropy losses $\mathcal{L}\_T^{j,k}$ of experts. It is used to optimize both the delegator and experts.
> >
> > **Q4:**  The delegator is 24MB and the expert is 110MB. How to get the accuracy of 100MB?
> >
> > **A4:**  This is achieved by early termination strategy and we have added its introduction in section 3.1 of the revision. The delegator also outputs a rough prediction, we will accept it if its Maximum Class Probability (MCP, probability of the predicted class) is larger than the threshold, otherwise, adopt the selected expert for a refined prediction. By varying this threshold, we can get a series of accuracies with averaged FLOPs/Instance from 24M to (24+110) M.
> >
> > **Q5:**  The early termination (early exit) is virtually not mentioned in the whole paper.
> >
> > **A5:**  Thank you for your comment which improves the completeness of this paper. We have revised our paper to improve the clarity of the inference procedure of CoE (early termination). We emphasize that delegator can trigger the potential early termination in ''abstract''. Then, briefly introduce the inference procedure in the second paragraph of ''introduction'' and demonstrate early termination is used to save FLOPs. Moreover, we have added a new section 3.1 to introduce the details of it in ''method''. In ''experiment'', we mention that the accuracy curves for CoE in Fig.4a are drawn by varying the threshold of early termination and state that the statistics of CoE in Table 1 are picked out from those curves.
> >
> > As for whether/when it is used, we regard it as a necessary component of CoE during inference as shown in Figure.1. Moreover, it is a pure inference strategy and has nothing to do with the training procedure.
> >
> > **Q6:**  Whether the code will be released upon acceptance. Given that some parts of the paper are hard to parse, that would be very helpful.
> >
> > **A6:**  We are glad to communicate with any reader who still feels confused about the revised paper via email, etc. Our code and models will also be released.
> >
> > **Q7:**  Does CoE negatively affect batching vs standard networks.
> >
> > **A7:**   Yes, CoE cannot conduct batch processing as easily as standard networks do. Given a set of images, CoE firstly uses delegator to obtain the rough prediction and determine the selected expert for each image as illustrated in section 3.1 of the revision. This procedure uses standard batching. Afterward, MCP of each rough prediction is calculated. For samples with MCP larger than a given threshold, the final recognition result is derived from the rough prediction (early termination). Other samples are partitioned into n groups based on which expert is selected, Then batch processing can be conducted within each group to obtain the refined prediction.
> >
> > **Q8:**  MobileNetV3-Large should be compared in Table 2.
> >
> > **A8:**   We have added MobileNetV3-Large in Table 2 of the revision.

---

> > > ### Author Response · Authors · 2021-11-22
> > > **Reply to Reviewer SUb5 (3/3)**
> > >
> > > **Q9:**  Ablation study for elements of the optimization, like KD, stochastic depth and auto-augment.
> > >
> > > **A9:**   We have added the ablation study for knowledge distillation (KD), auto-augment (AA) and stochastic depth (SD) in Appendix B.4.4 of the reversion. We adopt the CoE-Large setting and use 4 experts. We find KD extremely important for CoE, it may indicate CoE is easy to be overfitted. In addition, SD decreases the accuracy of CoE. By removing SD, CoE-Large (4 experts) boosts the accuracy from 79.9\% to 80.2\%. Perhaps, it is because SD makes the capacity of delegator and each expert too tiny [B]. By the way, these elements are widely-used strategies to overcome the overfitting problem. We think only when the overfitting problem is solved can task accuracy reflect model capacity exactly. Because this paper is concerned with improving model capacity with limited computation cost, we use them in the original paper.
> > >
> > > | KD| AA | SD     |Experts|FLOPs|TOP-1 Acc.|
> > > | :----:|    :----:   |          :----: |          :----: |          :----: |:----: |
> > > | ✓| ✓| ✓   |4|220M   |**79.9\%**|
> > > | ✓| ✓|    |4|220M   |**80.2\%**|
> > > | ✓| | ✓   |4|220M   |79.4\%|
> > > | | ✓| ✓   |4|220M   |76.2\%|
> > > | ✓| |   |4 |220M   |79.7\%|
> > > | | ✓| |4|220M   |**76.3\%**|
> > > | | | ✓  |4 |220M   |75.2\%|
> > > | | |    |4|220M   |75.1\%|
> > >
> > > **Q10:**  Ablation study for elements of the method, like WGM, LGM, SRM and progressive sharpening of the assignments.
> > >
> > > **A10:**   We have added the ablation study in appendix B.4.3 of the revision. CoE consists of 3 major components: WGM, LGM and SRM. Apart from directly removing each one component, we also try to alter some elements inside them. We propose several modified versions of CoE for ablation as below:
> > > * **CoE**$^{WGM}$ : Remove WGM from CoE. Thus, losses of experts have identical weights for each sample.
> > >
> > > * **CoE**$^{WGM^\star}$: WGM partitions the training data based on expert suitability, i.e. the assignment matrix $\displaystyle A_{m\times n}$ in WGM equals the output matrix $\displaystyle L_{m\times n}$ of LGM.
> > >
> > > * **CoE**$^{WGM^\circ}$: Remove the ''$\sum\_{j}A\_{j,k} =m/n$'' constraint in Eq.4, so that $\displaystyle A_{m\times n}$ neglects the **No Superiority Assumption (NSA)**. Then, take $A_{m\times n}$ as the output of WGM without the smoothing and normalizing(Eq.5&6 of the revision).
> > >
> > > * **CoE**$^{WGM^\bullet}$: Remove the progressive sharpening of assignment in WGM. Specifically, using a constant 0.8 for $\alpha$ in Eq.5, instead of linearly increasing it from 0.2 to 0.8.
> > >
> > > * **CoE**$^{LGM}$: Remove LGM from CoE. Thus, CoE collapse to a single expert with delegator to trigger the early termination.
> > >
> > > * **CoE**$^{LGM^\star}$: Abandon the refining of suitability metric (Eq.2) and remove the ''$\sum\_{j}L\_{j,k} =m/n$''constraint in Eq.3. So that $\displaystyle L_{m\times n}$ neglects the **No Superiority Assumption（NSA)**.
> > >
> > > * **CoE**$^{SRM}$: Remove SRM from CoE.
> > >
> > > | Method| Experts| FLOPs|TOP-1 Acc.|
> > > | :----:|    :----:   |          :----: |          :----: |
> > > | **CoE-Large**$^{WGM}$| 4|220M   |78.1\%|
> > > | **CoE-Large**$^{WGM^\star}$| 4|220M   |78.4\%|
> > > | **CoE-Large**$^{WGM^\circ}$| 4|220M   |79.2\%|
> > > | **CoE-Large**$^{WGM^\bullet}$|   4 |220M   |79.4\%|
> > > | **CoE-Large**$^{LGM}$| 4   |220M   |78.0\%|
> > > | **CoE-Large**$^{LGM^\star}$| 4   |220M   |79.4\%|
> > > | **CoE-Large**$^{SRM}$| 4   |220M   |79.5\%|
> > > | **CoE-Large**|   4  |220M   |79.9\%|
> > >
> > > We conduct experiments for those CoE versions with the **CoE-Large** setting and **4 experts**. Results are shown in the table above, and the conclusions are listed below:
> > >
> > > * WGM and LGM are the most important components in CoE. The removal of WGM and LGM reduce the accuracy from 79.9\% to 78.1\% and 78.0\%, respectively.
> > >
> > > *  In WGM, the training data should be partitioned based on delegator. Otherwise, delegator will overfit to irregular selection labels as illustrated in section 3.3. That is why **Coe-Large**$^{WGM^\star}$ only achieves an accuracy of 78.4\%.
> > >
> > > * The **No Superiority Assumption (NSA)** is importart for CoE. Without this assumption, **CoE-Large**$^{WGM^\circ}$ and **CoE-Large**$^{LGM^\star}$ only reach the accuracy of 79.2\% and 79.4\%.
> > >
> > > * The progressive sharpening of assignment in WGM can also boost the performance, thus the accuracy for **CoE-Large**$^{WGM^\bullet}$ is 0.5\% lower than **CoE-Large**.
> > >
> > > * The component SRM is also useful. It promotes delegator to select experts better, yielding a 0.4\% improvement for accuracy.
> > >
> > > ---
> > >
> > > [A] Devansh Arpit, et al. A closer look at memorization in deep networks. ICML, 2017.
> > >
> > > [B] Tradeoffs in data augmentation: An empirical study. ICLR 2021.
> > >
> > > ---

---

> > > > ### Comment · Reviewer_SUb5 · 2021-11-24
> > > > **Thank you for the reply**
> > > >
> > > > Thanks for the careful and detailed reply. I have read the rebuttal and also other author's comments and rebuttals.
> > > >
> > > > Regarding the rebuttal to my review, the authors have done a good job overall:
> > > > - The issue with proper ablations, which has been highlighted by other reviewers too, has been tackled very well.
> > > > - I understand that FLOPs are on average (mixing early exit and standard exit)... that should be clarified. Sometimes you need to work with worse-case-scenario metrics for example. It isn't a big problem, but something that should be clear to the reader
> > > > - I am happy to hear about the intention to release the code
> > > > - There has been an honest effort towards clarifying the roles of the model components. I think though part of the issue stems from the way the authors are explaining it, rather from the specific wording. Explanations are from the perspective of someone very involved in the work and very familiar with what each component of the system does. Thus, they can reference all these interplays between the components. This is harder to parse for a casual reader. I am not sure what the solution would be - some graphical depiction maybe? In any case, I think there's an improvement but it is not a solved problem.
> > > >
> > > > Regarding other comments:
> > > > - Clarity has been highlighted as an issue by several reviewers, and I think is the biggest issue with the current paper.
> > > > - Higher memory footprint is indeed a drawback. Similarly batching is not ideal. But having a strong issue with that is having an issue with not having a free lunch. More memory cost for higher performance is a very interesting trade-off in my opinion (and one I would pay almost any time!)
> > > > - While it is true that some aspects of the paper are not very clear, I like this kind of paper that puts a bunch of things together to build something remarkable. They have some good novelty of their own, but on top of that, they use all the tools on the toolbox to push the performance to a new level. Otherwise we will be seeing the same deltas over out-of-fashion optimization strategies over an over. Figure out how things can be combined to maximize the efficiency-accuracy trade-off return, and how well different technical contributions stack together (if they do at all), is an excellent contribution. If this happens on top of their own technical contribution, then this is a double kudos, not a minus in my opinion.
> > > > - 80% on ImageNet is not easy... especially with 100MFLOPs. One thing that is clear with this paper is that this figure is impressive.
> > > >
> > > > I was already positive about this paper. All in all, after the rebuttal I have more confidence that I would like to see it published.

---

### Official Review · Reviewer_dWQ7 · 2021-11-02

**Correctness:** 3
**Technical Novelty And Significance:** 3
**Empirical Novelty And Significance:** 3
**Recommendation:** 6
**Confidence:** 3

**Main Review:**

Strength:

1.	The idea of model collaboration is interesting and new.

2.	The model has achieved 80% accuracy within 100M FLOPS, achieving the SoTA performance on ImageNet.

Weakness:

1.	About related work.
The paper mentions that the delegator needs to select an expert from the candidate experts and compares to other model selection methods. However, the related work of model selection [A, B] is lacking in the paper. The authors are strongly suggested to enrich the related work part and compare with them.

[A] Ranking Neural Checkpoints. CVPR2021.

[B] LogME: Practical Assessment of Pre-trained Models for Transfer Learning. ICML2021.

2.	About experiments.
+ As the proposed method contains several networks (including delegator and 4/16 experts), which probably introduces larger memory cost. Can the authors also elaborate on the additional memory cost and compare it with other baseline models in the experimental part.
+ In Sec. 4.3.1, while the paper mainly targets the model ensemble problem, the authors only compare the method with one simple model ensemble method, which is not very convincing. Therefore, the authors are suggested to conduct a more extensive comparison with other model ensemble methods.


======================

Post-rebuttal:

I acknowledge that it for the first introduced an instance-wise model selection framework, which is novel. Most of my concerns about the insufficient experiments have been addressed by the rebuttal. I agree to upgrade the score to 6 and hope the authors can improve the writing and paper readability in the next version.

**Summary Of The Paper:**

In this paper, the author considers the model collaboration problem, specifically ensemble learning. The author identifies two issues of ensemble learning: significant runtime cost and high memory access cost. To alleviate these issues, the authors propose a model collaboration framework named Collaboration of Experts (CoE) along with a training algorithm designed for the framework. The training algorithm contains three components: weight generation module (WGM), label generation module (LGM), and selection reweighting module (SRM). Experiments are conducted on ImageNet to demonstrate the effectiveness of the proposed method.

**Summary Of The Review:**

The problem of this paper is interesting, but the experiments are not sufficient to support the claim. Please see the main review for the details.

---

> ### Author Response · Authors · 2021-11-22
> **Reply to Reviewer dWQ7 (1/2)**
>
> Thank you for your comments which improves the completeness of this paper. We have revised our paper based on the reviews, including the comparison to works about model selection in section 2.1 and the analysis of memory cost of CoE in section 4.2.2. We try to answer your questions as below.
>
> **Q1:** Compare with works about model selection, like [A, B].
>
> **A1:** We have added the comparison in section 2.1 of the revision. These methods select models (or checkpoints) task-wisely, while CoE selects models instance-wisely. They are concerned with ranking some pre-trained models and find the one transfers best to a downstream task of interest.  By contrast, CoE tries to improve task performance by selecting the most suitable expert for each instance. Moreover, these methods conduct model selection based on a set of samples (training set), thus cannot be adopted in instance-wise model selection.
>
> **Q2:** Compare the memory cost of CoE with other baseline models.
>
> **A2:** We elaborate on the memory cost of CoE in Table 2 and section 4.2.2 of the revision. We analyze the memory cost from two perspectives: the number of parameters and memory access cost (MAC). As can be seen, the accuracy of CoE-Large (4 experts) is no worse than BasisNet and CondConv-EfficientNet-B0 when using similar parameters. Besides, the averaged MAC/Instance of CoE is much smaller than theirs. Compared with GhostNet 1.3x, the accuracy for CoE-Large (16 experts) is 5.0\% higher with a smaller MAC.
>
>  | Method| FLOPs|MAC|Params|Top-1 Acc.|
> | :----:|    :----:   |          :----: |          :----: | :----: |
> | MobileNetV3-Small| 56M|2.5M|2.5M   |67.4%|
> | GhostNet 1.0x| 141M|5.2M|5.2M|73.9%|
> | TinyNet-B| 202M|   3.7M|3.7M|75.0%|
> | MobileNetV3-Large|   219M|5.4M|5.4M|75.2%|
> | GhostNet 1.3x| 226M|   7.3M|7.3M|75.7%|
> | OFA-230| 230M|   5.8M|5.8M|76.9%|
> | EfficientNet-B0| 391M|   5.3M|5.3M|77.2%|
> | TinyNet-A| 339M|   5.1M|5.1M|77.7%|
> | CondConv-EfficientNet-B0| 413M|   24.0M|24.0M|78.3%|
> | BasisNet| 198M|   24.9M|24.9M|80.0%|
> | CoE-Large (4 experts)| 220M|   6.6M|25.7M|79.9%|
> | CoE-Large (16 experts)| 194M|   6.0M|95.3M|80.7%|
>
> By the way, we have also analyzed the memory cost in section 4.4. Compared to Transformer (big), CoE-Transformer achieves similar performance with much smaller MAC and parameters.
>
>  | Model|MAC|Params|BLEU|
> | :----:|    :----:   |          :----: |          :----: |
> | Transformer (base model)| 62.4M|62.4M|28.1|
> | Transformer (big)| 213.0M|213.0M|29.3|
> | CoE-Transformer| 62.5M|138.2M|29.4|

---

> > ### Author Response · Authors · 2021-11-22
> > **Reply to Reviewer dWQ7 (2/2)**
> >
> > **Q3:** While this paper mainly targets the model ensemble problem, the authors only compare with one simple model ensemble method, which is not very convincing.
> >
> > **A3:**  Naive ensemble is simple, but its performance is competitive. As can be seen from the table below, naive ensemble has a competitive performance in terms of accuracy. Nonetheless, CoE-Large achieves much higher accuracy than it, with a gap of 1.3\%. It demonstrates the superiority of CoE in terms of accuracy. More importantly, CoE has a smaller computation cost compared with ensemble methods. The computation cost of any ensemble method is no smaller than the one of base model, but it is not the case for CoE thanks to the early termination strategy. CoE-Large$^\star$ (using a smaller threshold for early termination) can achieve 80.7\% accuracy (still 1.1\% higher than naive ensemble) with only $0.84\times$ FLOPs of the base model, thus CoE is superior in computation cost as well. MIMO[C] achieves remarkable performance with ResNet50 as the base model, but its performance drops a lot than naive ensemble when the base model is changed to a compact model (OFA-230). It is because MIMO is built on the ''heavy parameter redundancy'' assumption of the base model, while OFA-230 is compact enough and has few redundant parameters.
> >
> > The advantage of CoE comes from its efficient model collaboration scheme: **delegation** scheme, i.e. assigning one or two models conditionally to make the prediction. By contrast, ensemble methods are based on the **consensus** scheme [C, D, E, F, H]. This is also the core difference between CoE and ensemble methods. Considering conditional computation methods are built on this scheme, perhaps it is more proper that CoE mainly targets the conditional computation problem. Motivated by this, we compare CoE with many conditional computation methods [I, J, K, L] in the original paper.
> >
> >  | Method| Base Model |FLOPs |Top-1 Acc.|
> > | :----:|    :----:   |          :----: |          :----: |
> > |Complementary Ensemble[F]| ResNet50 | 4 $\times$ 4090M  |77.9\% [F]|
> > |Naive Ensemble| ResNet50 | 4$\times$4090M  |77.5\% [C]|
> > |BatchEnsemble[D]| ResNet50| 4$\times$4090M  |76.7\% [C]|
> > |Deep Boosting[H] | ResNet50 | 4$\times$4090M  |76.3\% [F]|
> > |MIMO[C]| ResNet50 | 1$\times$4090M  |77.5\% [C]|
> > |Naive mutlihead| ResNet50 | 1$\times$4090M  |76.6\% [C]|
> > |Naive Ensemble| OFA-230 | 4$\times$230M  |79.6\%|
> > |MIMO[C]| OFA-230 | 1$\times$230M  |77.7\%|
> > |CoE-Large$^\star$| OFA-230 | **0.84**$\times$230M  |80.7\%|
> > |CoE-Large| OFA-230 | 0.96$\times$230M  |**80.9\%**|
> >
> > -----
> >
> > [A] Ranking Neural Checkpoints. CVPR 2021.
> >
> > [B] LogME: Practical Assessment of Pre-trained Models for Transfer Learning. ICML 2021.
> >
> > [C] Training independent subnetworks for robust prediction. ICLR 2021.
> >
> > [D] BatchEnsemble: An alternative approach to efficient ensemble and lifelong learning. ICLR 2020.
> >
> > [E] Neural network ensembles. PAMI 1990.
> >
> > [F] Embedding Complementary Deep Networks for Image Classification. CVPR 2019.
> >
> > [H] Boosting neural networks. Neural Computation 2000.
> >
> > [I] Dynamic convolution: Attention over convolution kernels. CVPR 2020.
> >
> > [J] Multi-scale dense networks for resource efficient image classification. ICLR 2018.
> >
> > [K] Weightnet: Revisiting the design space of weight networks. ECCV 2020.
> >
> > [L] Conditionally parameterized convolutions for efficient inference. NIPs 2020.
> >
> > ---

---

> > > ### Comment · Reviewer_dWQ7 · 2021-11-25
> > > **Post-rebuttal**
> > >
> > > Thanks for the authors' responses, which addressed most of my concerns except for the first one.
> > >
> > > About the comparison with model selection methods [A, B], the authors claimed that they select model task-wisely while this work selects model instance-wisely. However, the advantages of model selection instance-wisely compared to task-wisely are not clear. Maybe some experimental comparisons can help.

---

> > > > ### Author Response · Authors · 2021-11-26
> > > > **Reply to Reviewer dWQ7**
> > > >
> > > > Thanks for your valuable and timely response. Those task-wise model selection methods [A, B] need at least **two tasks**. One is for training a number of models (called ’’pre-trained models’’), the other is the task of interest. The problem they study is how to select a pre-trained model that has the best performance when **transferred** to the task of interest. While only **one task** is involved in our studied problem, we are not sure how to get a series of pre-trained models, actually.  My best guess is using the given task to train some models then select the best one. This problem is what NAS methods [M, N, O, P, Q] are studied. NAS methods aim at searching for the best neural network architecture among a large search space on a given task. In section 4.2.1, we have compared with some outstanding NAS methods (e.g. MobileNetV3[M], OFA[N], TinyNet[O], EfficientNet[P], FBNetV3[Q]). The superiority of our proposed CoE (in Tabel 1 and Fig.4a) demonstrates the advantage of model selection instance-wisely compared to task-wisely. If our understanding for how to implement task-wise model selection when only one task is given is not satisfactory, could you elaborate more on  a  reasonable comparison? We would appreciate it very much!
> > > >
> > > > | Method | FLOPs | Top-1 Acc. |
> > > > | :----:| :----:| :----: |
> > > > | TinyNet-B[O] | 202 M| 75.0\% |
> > > > | MobileNetV3-Large[M] | 219 M| 75.2\% |
> > > > | OFA-230[N] | 230 M| 76.9\% |
> > > > | TinyNet-A[O] | 339 M| 77.7\% |
> > > > | OFA-595A[N] | 595 M| 80.0\% |
> > > > | EfficientNet-B2[P] | 1000 M| 80.1\% |
> > > > | FBNetN3-C[Q] | 557 M| 80.5\% |
> > > > | CoE-Small | 100 M| 78.2\% |
> > > > | CoE-Small+CondCov+PWLU | 100 M| 80.0\% |
> > > > | CoE-Large | 194 M| 80.7\% |
> > > > | CoE-Large+CondCov | 214 M| 81.5\% |
> > > >
> > > > By the way, we have added the comparison between [A, B]  and our proposed CoE in section 2.1 of the revision.
> > > >
> > > > ---
> > > >
> > > > [A] Ranking Neural Checkpoints. CVPR 2021.
> > > >
> > > > [B] LogME: Practical Assessment of Pre-trained Models for Transfer Learning. ICML 2021.
> > > >
> > > > [M] Searching for mobilenetv3. ICCV 2019.
> > > >
> > > > [N] Once-for-all: Train one network and specialize it for efficient deployment. ICLR 2020.
> > > >
> > > > [O] Model rubik’s cube: Twisting resolution, depth and width for tinynets. NeurIPS 2020.
> > > >
> > > > [P] Efficientnet: Rethinking model scaling for convolutional neural networks. ICML 2019.
> > > >
> > > > [Q] Fbnetv3: Joint architecture-recipe search using neural acquisition function. CoRR 2020.
> > > >
> > > > ---

---

> > > > > ### Author Response · Authors · 2021-11-29
> > > > > **Thank you again for your time and valuable comments**
> > > > >
> > > > > We would like to thank you again for your time and valuable comments, and hopefully our explanations could address your concerns. As we are nearing the end of the discussion period, do you still have any concerns? We are happy to address them.

---

### Author Response · Authors · 2021-11-21
**Revised manuscript is uploaded**

We thank all the reviewers for their valuable comments and detailed feedback. We have revised the manuscript according to the reviewers' comments to address concerns and to avoid any unnecessary confusion. Changes in the revised manuscript are highlighted in blue.

**For clarity**:
* Apart from improving the clarity for the components of CoE (i.e. WGM, LGM and SRM), we also try to provide some in-depth analysis about ''why they work?'' and ''why they are designed like this?''. We introduce ''what they aim to do'' in ''abstract''. Then, the third paragraph of ''introduction'' briefly illustrates ''what they do'' along with ''why they work''. In ''method'' (section 3.2, 3.3 and 3.4), we try to improve the clarity of our statements about their details. In the first three paragraphs of section 3.3, we analyze why delegator and experts learn to collaborate, in addition to the collapse caused by a different data partition manner (the one LGM adopts) in WGM with a demo Fig.3.

* Improving the clarity for the inference procedure of CoE (early termination). We emphasize that delegator can trigger the potential early termination in ''abstract''. Then, briefly introduce the inference procedure in the second paragraph of ''introduction'' and demonstrate early termination is used to save FLOPs. Moreover, we have added a new section 3.1 to introduce the details of it in ''method''. In ''experiment'', we mention that the accuracy curves for CoE in Fig.4a are drawn by varying the threshold of early termination and state that the statistics of CoE in Table 1 are picked out from those curves.

* Adding the comparison with works about model selection [A, B] in section 2.1 according to the suggestion of reviewer dWQ7.

* Improving the clarity by revising the English and the style of writing. For example, we have modified the references according to the suggestion of reviewer u2o7. “citep” is used and “citecolor” are set as violet to avoid confusion.

**For experiments**:
* Elaborated ablations are conducted to illustrate the importance of each element of CoE in Appendix B.4.3. The ablations aim to demonstrate the significance of each component (WGM, LGM and SRM), as well as the effect of ''No Superiority Assumption'' in WGM/LGM, the dataset partition manner  in WGM and the progressive sharpening of assignment in WGM.

*  Elaborated ablations are conducted for the training strategies (i.e. knowledge distillation, auto-augment and stochastic depth) in Appendix B.4.4.

* Experiments about the memory cost of CoE are added in section 4.2.2 according to the suggestion of reviewer dWQ7.

-----

[A] Ranking Neural Checkpoints. CVPR 2021.

[B] LogME: Practical Assessment of Pre-trained Models for Transfer Learning. ICML 2021.

-----

---

### Author Response · Authors · 2021-11-21
**The added ablation studies in revised manuscript**

Ablation study for each element of CoE
---
---
CoE consists of 3 major components: WGM, LGM and SRM. Apart from directly removing each one component, we also try to alter some elements inside them. We propose several modified versions of CoE for ablation as below:
* **CoE**$^{WGM}$ : Remove WGM from CoE. Thus, losses of experts have identical weights for each sample.

* **CoE**$^{WGM^\star}$: WGM partitions the training data based on expert suitability, i.e. the assignment matrix $\displaystyle A_{m\times n}$ in WGM equals the output matrix $\displaystyle L_{m\times n}$ of LGM.

* **CoE**$^{WGM^\circ}$: Remove the ''$\sum\_{j}A\_{j,k} =m/n$'' constraint in Eq.4, so that $\displaystyle A_{m\times n}$ neglects the **No Superiority Assumption (NSA)**. Then, take $A_{m\times n}$ as the output of WGM without the smoothing (Eq.5) and normalizing(Eq.6).

* **CoE**$^{WGM^\bullet}$: Remove the progressive sharpening of assignment in WGM. Specifically, using a constant 0.8 for $\alpha$ in Eq.5, instead of linearly increasing it from 0.2 to 0.8.

* **CoE**$^{LGM}$: Remove LGM from CoE. Thus, CoE collapse to a single expert with delegator to trigger the early termination.

* **CoE**$^{LGM^\star}$: Abandon the refining of suitability metric (Eq.2) and remove the ``$\sum\_{j}L\_{j,k} =m/n$''constraint in Eq.3. So that $\displaystyle L_{m\times n}$ neglects the **No Superiority Assumption（NSA)**.

* **CoE**$^{SRM}$: Remove SRM from CoE.

| Method| Experts| FLOPs|TOP-1 Acc.|
| :----:|    :----:   |          :----: |          :----: |
| **CoE-Large**$^{WGM}$| 4|220M   |78.1\%|
| **CoE-Large**$^{WGM^\star}$| 4|220M   |78.4\%|
| **CoE-Large**$^{WGM^\circ}$| 4|220M   |79.2\%|
| **CoE-Large**$^{WGM^\bullet}$|   4 |220M   |79.4\%|
| **CoE-Large**$^{LGM}$| 4   |220M   |78.0\%|
| **CoE-Large**$^{LGM^\star}$| 4   |220M   |79.4\%|
| **CoE-Large**$^{SRM}$| 4   |220M   |79.5\%|
| **CoE-Large**|   4  |220M   |79.9\%|

We conduct experiments for those CoE versions with the **CoE-Large** setting and **4 experts**. Results are shown in the table above, and the conclusions are listed below:

* WGM and LGM are the most important components in CoE. The removal of WGM and LGM reduce the accuracy from 79.9\% to 78.1\% and 78.0\%, respectively.

*  In WGM, the training data should be partitioned based on delegator. Otherwise, delegator will overfit to irregular selection labels as illustrated in section 3.3. That is why **Coe-Large**$^{WGM^\star}$ only achieves an accuracy of 78.4\%.

* The **No Superiority Assumption (NSA)** is importart for CoE. Without this assumption, **CoE-Large**$^{WGM^\circ}$ and **CoE-Large**$^{LGM^\star}$ only reach the accuracy of 79.2\% and 79.4\%.

* The progressive sharpening of assignment in WGM can also boost the performance, thus the accuracy for **CoE-Large**$^{WGM^\bullet}$ is 0.5\% lower than **CoE-Large**.

* The component SRM is also useful. It promotes delegator to select experts better, yielding a 0.4\% improvement for accuracy.


Ablation study for the training strategies
---
---
Knowledge distillation (KD), auto-augment (AA) and stochastic depth (SD) are widely-used strategies to overcome the overfitting problem. We think only when the overfitting problem is solved can task accuracy reflect model capacity exactly. Because this paper is concerned with improving model capacity with limited computation cost, we use these strategies. Nonetheless, we conduct ablations for them.  We adopt the **CoE-Large** setting and use **4 experts**. Results are shown in Table below. We find KD extremely important for CoE, it may indicate CoE is easy to be overfitted. In addition, SD decreases the accuracy of CoE. By removing SD,  **CoE-Large (4 experts)**  boosts the accuracy from 79.9\% to 80.2\%. Perhaps, it is because SD makes the capacity of delegator and each expert too tiny [A].

| KD| AA | SD     |Experts|FLOPs|TOP-1 Acc.|
| :----:|    :----:   |          :----: |          :----: |          :----: |:----: |
| ✓| ✓| ✓   |4|220M   |**79.9\%**|
| ✓| ✓|    |4|220M   |**80.2\%**|
| ✓| | ✓   |4|220M   |79.4\%|
| | ✓| ✓   |4|220M   |76.2\%|
| ✓| |   |4 |220M   |79.7\%|
| | ✓| |4|220M   |**76.3\%**|
| | | ✓  |4 |220M   |75.2\%|
| | |    |4|220M   |75.1\%|

---

[A] Tradeoffs in Data Augmentation: An Empirical Study. ICLR 2021

---

---

### Author Response · Authors · 2021-11-24
**Thank all the reviewers for their great efforts in reviewing our paper!**

We thank all the reviewers for their great efforts in reviewing our paper! We are looking forward to knowing whether our response has well addressed your concerns. Please feel free to leave your comments if you have any further questions.

---

### Decision · Program_Chairs · 2022-01-20

**Decision:**

Reject

**Comment:**

This paper trains an expert style DNN that routes input examples to appropriate expert modules resulting in high accuracy on ImageNet with less compute. Reviewers have been positive about the strong empirical results. However the paper itself is not written well and reviewers had hard time figuring out actual architecture and training methodology. For example reviewers couldn't easily figure out the differences between LGM, WGM and SRM.

The paper itself is sparse on why some of the choices have been made, their relation to existing methods and how do they affect the final performance. For example - In eq2, TCP objective has been normalized for each expert separately with a vague No Superiority Assumption. What motivates this assumption? Why is it reasonable? Eq 4 is quite similar to the load balancing loss in Switch Transformer paper. However there has been no discussion about the similarities and differences.

I think the paper needs to rewritten with clear explanation of the actual architecture, in what aspects it is similar/differs to existing expert models. What key components are the reason for the superior performance?

While I appreciate the authors for the ablations studies they presented during response phase, I think the paper requires major rewriting and cannot recommend acceptance at this stage.